# Thermodynamic properties of isoprene and monoterpene derived organosulfates estimated with COSMO*therm*

Noora Hyttinen[1], Jonas Elm[2], Jussi Malila[1], Silvia M. Calderón[1], and Nønne L. Prisle[1]

[1]Nano and Molecular Systems Research Unit, University of Oulu, P.O. Box 3000, 90014 Oulu, Finland
[2]Department of Chemistry and iClimate, Aarhus University, Langelandsgade 140, 8000 Aarhus C, Denmark

**Correspondence:** Noora Hyttinen (noora.hyttinen@oulu.fi), Nønne L. Prisle (nonne.prisle@oulu.fi)

**Abstract.** Organosulfates make significant contributions to atmospheric secondary organic aerosol (SOA), but little is still known about the thermodynamic properties of atmospherically relevant organosulfates. We have used the COSMO*therm* program to calculate both gas- and condensed-phase properties of previously identified atmospherically relevant monoterpene and isoprene derived organosulfates. Properties include solubilities, activities and saturation vapor pressures, which are critical to the aerosol phase stability and atmospheric impact of organosulfate SOA. Based on the estimated saturation vapor pressures, the organosulfates of this study can all be categorized as semi- or low-volatile, with saturation vapor pressures 4 to 8 orders of magnitude lower than that of sulfuric acid. The estimated p$K_a$ values of all the organosulfates indicate a high degree of dissociation in water, leading in turn to high dissociation corrected solubilities. In aqueous mixtures with inorganic sulfate, COSMO*therm* predicts a salting out of both the organosulfates and their sodium salts from inorganic co-solutes. The salting-out effect of ammonium sulfate (less acidic) is stronger than of ammonium bisulfate (more acidic). Finally, COSMO*therm* predicts liquid–liquid phase separation in systems containing water and monoterpene derived organosulfates. The COSMO*therm* estimated properties support the observed stability of organosulfates as SOA constituents and their long range transport in the atmosphere, but also show significant variation between specific compounds and ambient conditions.

## 1 Introduction

Organosulfates ($R-OSO_3H$, OS) have been identified as components of atmospheric secondary organic aerosol (SOA) from a variety of environments (Surratt et al., 2007; Glasius et al., 2018a, b). In the Amazon, the contribution of organic sulfate was found to be 3–42% of the total aerosol sulfate for the compounds measured using aerosol mass spectrometry (Glasius et al., 2018a). In Atlanta, Georgia, organosulfates accounted for 16.5% of the total organic carbon of fine particulate matter ($PM_{2.5}$) (Hettiyadura et al., 2019).

Multiple laboratory studies have shown that organosulfates are formed in the condensed phase from reactions between organic molecules and either a sulfate ion ($SO_4^{2-}$) (Iinuma et al., 2009; Minerath and Elrod, 2009) or a sulfate radical ($SO_4^{\cdot -}$) (Schindelka et al., 2013; Wach et al., 2019). Organosulfates have been seen to form, for instance, from oxidation products of monoterpenes (Surratt et al., 2008) and pinonaldehyde (Liggio and Li, 2006) in the presence of acidified sulfate seed and from isoprene derived organosulfates in the presence of sulfate (Darer et al., 2011). Some studies have suggested that the

formation of organosulfate correlates with the acidity of the aerosol particles (Chan et al., 2011) where more dilute acidic sulfate aerosol leads to a lower reactive uptake of isoprene epoxydiol ($C_5H_{10}O_3$, IEPOX) (Zhang et al., 2018), while other studies suggest that the abundance of the formed organosulfates correlates only with the sulfate content in the aerosol (Xu et al., 2015; Budisulistiorini et al., 2015).

Recent measurements close to Beijing using a Filter Inlet for Gas and Aerosol (FIGAERO) chemical ionization mass spectrometer (CIMS) have shown that sulfur containing organic compounds, such as organo/nitrooxy organosulfates and sulfonates, can also be present in the gas-phase (Le Breton et al., 2018). Higher temperatures promote the presence of sulfur compounds in the gas phase and furthermore the partitioning to the particle phase was found to be dependent on ambient relative humidity. In urban areas, such as Xi'an in northwestern China, organosulfates are primarily of anthropogenic origin (Huang et al., 2018), but already in semi rural locations 40 km northwest of Beijing, up to 19% of the sulfur containing organics have been identified to be of biogenic origin (Le Breton et al., 2018).

Very little is still known of the physico-chemical properties of specific atmospherically relevant organosulfates and how they affect the properties of SOA. This is in part due to challenges related to sampling and isolating sufficient amounts of organosulfate material from atmospheric organic aerosol for subsequent analysis of single component properties, as well as synthesizing adequate amounts of known organosulfate reference compounds. The hygroscopic properties of organosulfate containing aerosol have been measured using sodium salts of alkane sulfates (Woods III et al., 2007; Estillore et al., 2016) and limonene derived organosulfates (Hansen et al., 2015). Limonene derived organosulfate was demonstrated to lower the surface tension of aqueous solutions even more effectively than atmospherically relevant strong organic acids (Hansen et al., 2015). The effect of surface activity was evident in both sub-saturated hygroscopic growth and measured cloud condensation nuclei (CCN) properties of limonene derived organosulfate and its mixtures with ammonium sulfate (Hansen et al., 2015). Addition of organosulfates lowers the relative humidity of deliquescence and efflorescence transitions of sodium chloride aerosol (Estillore et al., 2016). In addition, Nguyen et al. (2014a, b) have seen indications of long-range transport of organosulfates, suggesting that organosulfates must have sufficiently low volatilities to remain in the aerosol phase over a wide range of atmospheric conditions.

In this study, we use the COSMO*therm* program to estimate different thermodynamic properties related to gas/condensed phase equilibrium of organosulfates and IEPOX in both pure water and aqueous mixtures with ammonium sulfate (($NH_4$)$_2SO_4$, AS) and bisulfate ($NH_4HSO_4$, ABS). The accuracy of COSMO*therm* p$K_a$ calculations (parametrization BP_TZVPD_FINE_C30_1601) is 0.65 log units RMSD (Klamt et al., 2016) and experimental saturation vapor pressures can be predicted within a factor of 2 with a tendency to overpredict experimental values of carboxylic acids (Schröder et al., 2016). For citric acid, using only conformers containing no intramolecular H-bonds leads to good agreement between experiments and COSMO*therm* estimated aqueous solubility, activity coefficients, density and p$K_a$ (Hyttinen and Prisle, 2020).

Figs 1 and 2 show the monoterpene and isoprene derived organosulfates, respectively, studied here. These compounds have previously been identified as components atmospheric aerosol (Surratt et al., 2007, 2008, 2010; Iinuma et al., 2009; Hansen et al., 2015). $\alpha$-Pinene-OS-1 and -2, and all of the $\beta$-pinene and limonene derived organosulfates are products of the monoterpene + OH reaction. $\alpha$-Pinene-OS-3 is formed from pinonaldehyde, $\alpha$-pinene-OS-4 from an oxidation product of $\alpha$-

pinene + OH, and $\alpha$-pinene-OS-5 and -6 are derived from pinonic acid. Isoprene-OS-1 and -2 are proposed to be formed from the aldehyde/keto form of an isoprene OH oxidation product in low-$NO_x$ conditions. Isoprene-OS-3 and -4 are likely formed from a nucleophilic attack by sulfate on the epoxy group of IEPOX (Darer et al., 2011). In field measurements in the US (Hettiyadura et al., 2017, 2019), an organosulfate corresponding to the chemical formula of isoprene-OS-3 and -4 dominated the bisulfate mass of $PM_{2.5}$. Since primary organosulfates are more stable against hydrolysis than tertiary organosulfates (Darer

et al., 2011), isoprene-OS-3 is likely the more abundant isomer, compared to isoprene-OS-4, in acidic aerosol.

For comparison to the monoterpene and isoprene derived organosulfate, we also studied the atmospherically abundant IEPOX ($C_5H_{10}O_3$, see the different isomers in Fig. S1 of the Supplement) and the smallest organosulfate, methyl bisulfate ($CH_3OSO_3H$).

## 2   Computational methods

We used COSMO*therm* release 19 (COSMO*therm*, 2019) to estimate several thermodynamic properties, such as acidity ($pK_a$), Henry's law solubility, activity and vapor pressure. The COSMO*therm* program is based on the conductor-like screening model for real solvents (COSMO-RS (Klamt, 1995; Klamt et al., 1998; Eckert and Klamt, 2002)). COSMO*therm* combines quantum chemistry and statistical thermodynamics to predict condensed-phase properties of liquids as well as partitioning between condensed and gas phases. Quantum chemical calculations provide input files (cosmo-files) for COSMO*therm* and the same

files can be used to estimate properties in various solutions. In addition, multiple conformers can be included in COSMO*therm* calculations to improve the description of conformer distributions in different solutions. Below we explain in detail how the input files for the COSMO*therm* calculations were computed and give definitions used by COSMO*therm* to estimate each of the properties. More detailed explanations for all of the methods can be found in the COSMO*therm* Reference manual (Eckert and Klamt, 2019). Without experimental reference data, we are not able to estimate the error for individual compounds. The

error estimates are same for all studied compounds and we therefore are not showing error bars in the figures.

### 2.1   COSMO input file generation

To generate the input files for the COSMO*therm* calculations, we used the COSMO*conf* program version 4.3 (COSMO*conf*, 2013). COSMO*conf* contains conformer generation algorithms, different levels of theory of quantum chemical calculations both for the condensed and the gas phase, and various methods for reducing the number of conformers in a way that does not

compromise the accuracy of the COSMO*therm* calculations.

Including multiple conformers in the COSMO*therm* calculations is important when the conformers have different polarities, as is the case for molecules that are able to form intramolecular hydrogen bonds (Eckert and Klamt, 2019). For finding an initial set of conformers, COSMO*conf* uses various conformer generating algorithms. However, none of these methods allow for the systematic conformer sampling of the molecules. The non-systematic conformer generation in COSMO*conf* has been

shown to lead to significantly different results in COSMO*therm* depending on the initial geometry with molecules containing hydroxy and hydroperoxy functional groups (Kurtén et al., 2018). Based on the recommendation by Kurtén et al. (2018), we

therefore used the systematic conformer sampling with the MMFF force fields in the Spartan '14 program (Wavefunction Inc., 2014). In addition to the most common carbon and oxygen atom types, MMFF force field is parametrized for the atom types of a sulfate group sulfur and oxygens (Halgren, 1996). This ensures that all unique conformers are found using the systematic sampling.

The conformers from Spartan '14 were used as input to COSMO*conf* where the TURBOMOLE program package version 7.11 (TURBOMOLE, 2010) was used for the quantum chemical calculations. Our calculation template in COSMO*conf* follows the BP-TZVPD-FINE-COSMO.xml template found in the program, omitting the conformational sampling step at the beginning and setting the cut-offs (energy and number of conformers based) of conformers high enough that no conformers were discarded. The gas-phase conformers were obtained by a BP/def-TZVP gas-phase geometry optimizations and BP/def2-TZVPD single-point energy calculations of the condensed phase geometries from COSMO*conf* using the *calculate* function in TURBOMOLE. The BP/def2-TZVPD-FINE//BP/def-TZVP level .cosmo and .energy files from COSMO*conf* and TURBO-MOLE were used in COSMO*therm* calculations. In addition, .cosmo, .energy and .vap files for $H_2O$ and the inorganic ions were taken from the COSMO*base*17 database (COSMO*base*, 2011).

## 2.2   COSMO*therm* calculations

In our COSMO*therm* calculations, we have used the most recent BP_TZVPD_FINE_19 parametrization. All calculations were done at 298.15 K. To the best of our knowledge, experimental information on the pure component phase state of most atmospherically relevant organics is not available. We therefore assume that all of the organosulfates (OS) and the isoprene epoxydiols (IEPOX) are liquid at 298.15 K. Without melting point and heat of fusion data, we are not able to accurately estimate the solubilities of solid-phase organosulfates. If the OS and IEPOX are solid at 298.15 K, the solubility results shown here are the mole fractions of the virtual liquid of the solute in the two liquid phases of a solid-liquid-liquid equilibrium. Sodium salts of the organosulfates ($R-OSO_3Na$, NaOS) are similar to sodium dodecyl sulfate (SDS) with regard to molar mass and functionality. SDS is solid at 298.15 K and we therefore assume that the NaOS are solid at this temperature. The organic compounds are treated as solutes and the aqueous solutions (pure water or binary aqueous ammonium salt mixtures) as the solvent.

To select the maximum number of conformers needed for COSMO*therm* calculations, convergence on the number of conformers was tested by calculating activities of isoprene-OS-1. In these test calculations, the change in activity of isoprene-OS-1 and $H_2O$ (in different mole fractions of isoprene-OS-1 in water) was at most 0.005 between 40 and 45 conformers of isoprene-OS-1. Based on this, the maximum number of conformers was set to 40 for larger monoterpene derived organosulfates and 50 for the smaller isoprene derived molecules. In COSMO*therm* calculations, conformers are weighted according to the Boltzmann distribution based on the sum of their solvated energy and chemical potential in the solution. However, normally only conformers with lowest solvated energies are selected for COSMO*therm* calculations. If the total number of unique conformers is high, not all conformers can be included in the COSMO*therm* calculation. When only a fraction of all conformers is used in a COSMO*therm* calculation, only those containing intramolecular H-bonds are used, as they have the lowest solvated energies. However, the interaction between a compound and water is more favorable for conformers containing no intramolecular

H-bonds. Therefore, in aqueous solutions, the chemical potential of conformers containing no intramolecular H-bonds is much lower than of conformers that contain multiple H-bonds (Hyttinen and Prisle, 2020). If a compound contains more unique conformers than can be included in COSMO*therm* calculations, more attention should be paid to selecting the conformers to represent the conformer distribution in the studied solutions.

Kurtén et al. (2018) found that COSMO*therm* (release 18, COSMO*therm* (2018)) overestimates the effect of intramolecular hydrogen bonds and recommended that only conformers containing no intramolecular hydrogen bonds should be used in saturation vapor pressure calculations. We tested the difference in saturation vapor pressures calculated using releases 18 and 19 (parametrizations BP_TZVPD_FINE_18 and BP_TZVPD_FINE_19, respectively) and found that differences between the two parametrizations are larger using all conformers than when only conformers containing no intramolecular H-bonds are

used. Variation between estimates using different conformer sets is also smaller in release 19 than in release 18. Hyttinen and Prisle (2020) found a good agreement between experimental and COSMO*therm* (release 19) estimated solubilities and activity coefficients when conformers containing intramolecular H-bonds were omitted from the COSMO*therm* calculations. We therefore omitted all conformers containing intramolecular hydrogen bonds from the calculations of OS and IEPOX. Generally, the omission of conformers containing intramolecular H-bonds leads to lower saturation vapor pressures (Kurtén

et al., 2018), higher aqueous solubility and larger deviation from ideality of activity coefficients (Hyttinen and Prisle, 2020).

The number of intramolecular H-bonds in the condensed phase was determined using release 18 of COSMO*therm*. For isoprene-OS-3 and isoprene-OS-4, we only found 2 and 0 conformers containing no hydrogen bonds, respectively. For these two species, we used all conformers containing no full and any number of partial intramolecular H-bonds or one full and no partial intramolacular H-bonds. Many of the deprotonated organosulfates (sodium salt anions) have only conformers that

contain intramolecular H-bonds. For this reason, we chose to use all of their lowest energy conformers in the COSMO*therm* calculations involving the NaOS. In calculation of $pK_a$ we used all conformers, since the calculation uses both the neutral and the ionic species.

### 2.2.1    Chemical potential

The chemical potential ($\mu$) of a component $i$ in a mixture is defined with respect to the chemical potential in a given reference

state $\mu_i^\circ$ with constant temperature $T$ and pressure $P$ as

$$\mu_i(x_i) = \mu_i^\circ(T, P) + RT \ln a_i, \tag{1}$$

where $R$ is the gas constant and $a_i = a_i(x_i)$ is the activity of component $i$ at a given actual mole fraction $x_i$, with respect to the chosen reference state. COSMO*therm* uses the pseudo-chemical potential (Ben-Naim, 1987) $\mu_i^*$, which is defined as

$$\mu_i^*(x_i) = \mu_i^\circ(T, P) + RT \ln \gamma_i, \tag{2}$$

where $\gamma_i$ ($= a_i/x_i$) is the activity coefficient of component $i$ at mole fraction $x_i$. By definition, the activity coefficient is 1 when component $i$ is in the reference state ($\gamma_i(x_i^\circ) = 1$). This means that in the reference state, chemical potential and pseudo-

chemical potential are equal:

$$\mu_i^{*\circ}(T,P) = \mu_i^{\circ}(T,P) \tag{3}$$

In COSMO*therm*, the pseudo-chemical potential of component $i$ in system $S$ is calculated using the $\sigma$-potential:

$$\mu_i^* = \mu_i^{C,S} + \int p_i(\sigma)\mu_S(\sigma)d\sigma, \tag{4}$$

where $p_i(\sigma)$ is the un-normalized $\sigma$-profile, and $\mu_S(\sigma)$ is the chemical potential of a surface segment with the screening charge density $\sigma$ (the $\sigma$-potential), which describes the affinity of the system $S$ to a surface of screening charge density $\sigma$. The combinatorial contribution to the chemical potential ($\mu_i^{C,S}$),

$$\mu_i^{C,S} = RT[(\hat{\lambda}_0 - \hat{\lambda}_1)\ln(r_i) + \hat{\lambda}_1(1 - \frac{r_i}{\bar{r}} + \ln\frac{r_i}{\bar{r}}) + \hat{\lambda}_2(1 - \frac{q_i}{\bar{q}} + \ln\frac{q_i}{\bar{q}}) - \hat{\lambda}_3\ln(r_i)], \tag{5}$$

is derived from the similar combinatorial free energy expression. The prefactors $\hat{\lambda}_0$, $\hat{\lambda}_1$ and $\hat{\lambda}_2$ have fixed values, while $\hat{\lambda}_3$ is adjustable. The total volume ($\bar{r}$) and area ($\bar{q}$) of all components $i$ are calculated as the mole fraction weighted sums of the dimensionless molecular volume ($r_i$) and area ($q_i$) of component $i$, respectively.

### 2.2.2 Activity coefficient

The activity coefficient of component $i$ at mole fraction $x_i$ in a mixture can be calculated using Eq. (2) as

$$\ln(\gamma_i(x_i)) = \frac{\mu_i^*(x_i) - \mu_i^{\circ}(T,P)}{RT} \tag{6}$$

The value of the activity coefficient in a given solution state $\{x_i\}$ depends on the choice of reference state. As the default reference state, COSMO*therm* uses the pure compound ($x_i^{\circ} = 1$, labeled as convention I (Levine, 2009) in the following) at $10^5$ Pa pressure and 298.15 K temperature. According to Eq. (3), with respect to this reference state, the pseudo-chemical potential is equal to the chemical potential when the system is in the reference state, $\mu_i^{*\circ,I}(x_i = 1) = \mu_i^{\circ,I}(x_i = 1)$, giving

$$\ln(\gamma_i^I(x_i)) = \frac{\mu_i^*(x_i) - \mu_i^{*\circ,I}(T,P)}{RT} \tag{7}$$

Activity coefficient values derived from experiments are often determined with respect to an ideal infinite dilution reference state ($x_i^{\circ} \to 0$, labeled as convention II (Levine, 2009)). For comparison with such experimentally derived values, activity coefficients for a given actual state $\{x_i\}$ determined with respect to the pure component reference state ($\gamma^I$) can be converted to the infinite dilution reference state ($\gamma^{II}$) as:

$$\ln\frac{\gamma_i^I(x_i)}{\gamma_i^I(x_i \to 0)} = \ln\gamma_i^I(x_i) - \ln\gamma_i^I(x_i \to 0)$$

$$= \frac{\mu_i^*(x_i) - \mu_i^{*\circ,I}(T,P)}{RT} - \frac{\mu_i^*(x_i = 0) - \mu_i^{*\circ,I}(T,P)}{RT}$$

$$= \frac{\mu_i^*(x_i) - \mu_i^*(x_i = 0)}{RT}$$

$$= \frac{\mu_i^*(x_i) - \mu_i^{*\circ,\mathrm{II}}(T,P)}{RT}$$

$$= \ln\gamma_i^{\mathrm{II}}(x_i), \tag{8}$$

where $\mu_i^*(x_i = 0) = \mu_i^{*\circ,\mathrm{II}}(T,P)$ follows from Eq. (2), since $\gamma^{\mathrm{II}} = 1$ at the reference state ($x_i^\circ \to 0$).

To the best of our knowledge, no experimental data on the isoprene and monoterpene derived organosulfates is currently available. Here, we are therefore not showing activity coefficients for these compounds with respect to infinite dilution reference state, but they can be calculated from our data using Eq. (8).

### 2.2.3 Solubility

We here calculate both absolute and relative solubilities of organosulfate solutes. The absolute solubilities are estimated by finding the liquid–liquid equilibrium (LLE, for liquid solutes) or the solid–liquid equilibrium (SLE, for solid solutes) using the solid–liquid equilibrium solver (SLESOL) in COSMO*therm*. For liquid solutes, the SLESOL finds the LLE between two phases ($\alpha$ and $\beta$) using the liquid phase equilibrium condition:

$$a_i^{\mathrm{I},\alpha} = a_i^{\mathrm{I},\beta} \tag{9}$$

In the LLE, Eq. (9) is true for both the solute and the solvent. Equation (9) is equivalent to the chemical potential of the solute being equal at the solubility limit in both phases, as opposed to the definition of the solubility of a solid solute, where the chemical potential of the solute at the solubility limit is equal to its chemical potential in the pure solute.

Based on their molecular structures, we expect organosulfates to have Brønsted acid properties. The acidity, in terms of the acid constant $\mathrm{p}K_\mathrm{a}$ ($= -\log K_\mathrm{a}$ for the equilibrium constant $K_\mathrm{a}$ corresponding to the equilibrium $\mathrm{R-OSO_3H + H_2O} \rightleftharpoons \mathrm{R-OSO_3^- + H_3O^+}$), is estimated using the deprotonated organosulfate species. COSMO*therm* estimates the $\mathrm{p}K_\mathrm{a}$ of compound $i$ from the molar free energy ($G$ in $\mathrm{kJ\,mol^{-1}}$) of the neutral and ionic species at infinite dilution using the linear free energy relationship (LFER):

$$\mathrm{p}K_\mathrm{a}^i = c + d(G_i^{\mathrm{anion}} - G_i^{\mathrm{neutral}}) \tag{10}$$

The LFER parameters for solvent water ($c = -130.152$ and $d = 0.116\ \mathrm{mol\,kJ^{-1}}$) are taken from COSMO*therm*'s parameter file. The energy difference ($G_i^{\mathrm{anion}} - G_i^{\mathrm{neutral}}$) is always positive, because in a neutral solvent, a neutral compound is more favorable than a charged compound. Relatively lower anion energy (more favorable deprotonation) leads to smaller energy difference leading to lower $\mathrm{p}K_\mathrm{a}$. The parametrization in COSMO*therm* currently enables calculation of $\mathrm{p}K_\mathrm{a}$ only in water, dimethylsulfoxide, acetonitrile or heptane. We are therefore not able to estimate $\mathrm{p}K_\mathrm{a}$ values of the organosulfates in other solvents relevant to this work, specifically aqueous ammonium sulfate and bisulfate solutions.

Dissociation in aqueous solution is expected to enhance solubility compared to the un-dissociated species. We use $\mathrm{p}K_\mathrm{a}$ values to calculate a dissociation correction to solubilities. The molar concentration of acid anion (A$^-$) after dissociation is

calculated using the pH of the solvent ($pH = 7.0$ for water) and $pK_a$ for the solute:

$$c_i^{A^-} = -0.5 \cdot 10^{-pH} + \sqrt{0.25 \cdot 10^{-2pH} + c_i^{HA} 10^{-pK_a}} \tag{11}$$

Here, the molar concentration of dissolved un-dissociated molecular organosulfate (HA) is calculated from the solubility mole fraction estimated using the SLESOL method, the mole fraction weighted density ($\rho$) of the system and the average molar mass of the solution ($M_{\text{solution}} = \sum_i x_i M_i$, where $M_i$ is the molar mass of component $i$):

$$c_i^{HA} = x_i \frac{\rho}{M_{\text{solution}}} \tag{12}$$

The calculation of composition-dependent solution densities is explained in Sect. S1 of the Supplement. The dissociation corrected mole fraction solubility ($x^{DC}$) is then calculated from the sum of the anionic and molecular molar concentrations using Eq. (12):

$$x_i^{DC} = (c_i^{HA} + c_i^{A^-}) \frac{M_{\text{solution}}}{\rho} \tag{13}$$

The average molar mass and composition-weighted density of the solution can be expressed using the mole fraction of the organic compound (see Sect. S1 of the Supplement for the equations), which is calculated iteratively from the dissociation corrected molar concentration $c_i^{HA} + c_i^{A^-}$.

For solid solutes, here the organosulfate sodium salts, the SLESOL finds the solid–liquid equilibrium (SLE) using the solid–liquid phase equilibrium condition:

$$\log_{10}(x_{\text{SOL},i}) = \frac{\mu_i^{*\circ,I} - \mu_i^*(x_i) - \Delta G_{\text{fus}}(T)}{RT \ln(10)} \tag{14}$$

The temperature-dependent molar free energy of fusion ($\Delta G_{\text{fus}} > 0\,\text{kJ mol}^{-1}$ for solid solutes) is an experimentally determined parameter, which can also be calculated from experimental molar heat of fusion ($\Delta H_{\text{fus}}$) and melting temperature ($T_{\text{melt}}$) using the Schröder–van Laar equation (Prigogine and Defay, 1954):

$$\Delta G_{\text{fus}}(T) = \Delta H_{\text{fus}}(1 - \frac{T}{T_{\text{melt}}}) - \Delta C_{p,\text{fus}}(T_{\text{melt}} - T) + \Delta C_{p,\text{fus}} T \ln \frac{T_{\text{melt}}}{T} \tag{15}$$

The heat capacity of fusion ($\Delta C_{p,\text{fus}}$) can be obtained from experiments, estimated as

$$\Delta C_{p,\text{fus}} = \frac{\Delta H_{\text{fus}}}{T_{\text{melt}}}, \tag{16}$$

or assumed to be zero. Equation (16) is physically a better estimate than $\Delta C_{p,\text{fus}} = 0\,\text{kJ mol}^{-1}\,\text{K}^{-1}$ for non-spherical and neutral compounds at temperatures above 150 K and within 200 K of the melting point (Eckert and Klamt, 2019). Since experimental data is not available for the organosulfate sodium salts, we here use the COSMO*therm* estimate of $\Delta C_{p,\text{fus}}$ in solubility calculations for solid solutes. As the melting point and heat of fusion, we use the experimental values of a related organosulfate compound, SDS, $T_{\text{melt}} = 478.15\,\text{K}$ (Rumble, 2018) and $\Delta H_{\text{fus}} = 50\,\text{kJ mol}^{-1}$ (heat of fusion of hydrated solid surfactant to micellar state (Shinoda et al., 1966)).

In COSMO*therm*, small atomic metal ions have extreme screening charge densities ($\sigma < $ -0.025 e Å$^{-2}$ or $\sigma > $ 0.025 e Å$^{-2}$). In reality, extreme screening charge densities of ions would lead to the formation of a solvation shell, where polar solvent molecules form strong H-bonds with the ion. This is not accounted for in COSMO*therm*, which leads to unrealistic behavior of the sodium ion in water. To improve the description of sodium cation solvation in case of the organosulfate sodium salts, we use a hydrated sodium cation instead of the dry sodium cation. Hydration of ions has previously been used in a model combining COSMO*therm* to describe the short range ion–molecule and molecule–molecule interactions, in combination with the Pitzer–Debye–Hückel solvation model (PDHS) to describe long range ion–ion interactions (Toure et al., 2014). The choice of hydration number for sodium is explained in more detail in Sect. S2 and Fig. S2 of the Supplement. The screening charge densities of larger ions, such as ammonium, sulfate and bisulfate, are less extreme (-0.025 e Å$^{-2} < \sigma < $ 0.025 e Å$^{-2}$, see Fig. S4 of the Supplement) and the non-hydrated ions can be used in COSMO*therm* calculations.

We also calculate solubilities in ternary systems containing water, organosulfate (OS or NaOS) and inorganic salt (($NH_4)_2SO_4$ or $NH_4HSO_4$). In these cases, the inorganic salt is considered part of the solvent and here treated in the form of its individual dissociated ions, leading to differently scaled mole fractions. Conversion of results from COSMO*therm*'s framework to the ternary system framework is explained in Sect. S2 and Fig. S3 of the Supplement.

Relative organic solubilities with respect to either the binary water–organic system or the ternary water–organic–inorganic salt system, are calculated using the relative screening option in COSMO*therm*. The relative solubilities are estimated using a zeroth order approximation of the solubility ($x^{(0)}_{\mathrm{SOL},i}$):

$$\log_{10}(x^{(0)}_{\mathrm{SOL},i}) = \frac{\mu^{*\circ,\mathrm{I}}_i - \mu^*_i(x_i = 0) - \max(0, \Delta G_{\mathrm{fus}}(T))}{RT \ln(10)} \tag{17}$$

where the solubility of component $i$ (in our case OS or NaOS) is assumed to be small enough to consider the component in a state of infinite dilution ($x_i = 0$) instead of the actual composition at the solubility limit ($x_i = x_{\mathrm{SOL}}$). In this approximation, the concentration of solute in the solvent is therefore assumed to be very small. The advantage of this zeroth order approximation in solubility calculation of solid solutes is that the solubility is calculated using only the chemical potential of the solute in the infinite dilution of the solvent, while the reference state (pure solute) chemical potential and the free energy of fusion cancel out. For a solute $i$ in two different systems with solvents $S1$ and $S2$:

$$\log_{10}(x^{S1,(0)}_{\mathrm{SOL},i}) - \log_{10}(x^{S2,(0)}_{\mathrm{SOL},i})$$

$$= \frac{\mu^{*\circ,\mathrm{I}}_i - \mu^{*,S1}_i(x_i = 0) - \Delta G_{\mathrm{fus}}(T)}{RT \ln(10)} - \frac{\mu^{*\circ,\mathrm{I}}_i - \mu^{*,S2}_i(x_i = 0) - \Delta G_{\mathrm{fus}}(T)}{RT \ln(10)}$$

$$= \frac{-\mu^{*,S1}_i(x_i = 0) + \mu^{*,S2}_i(x_i = 0)}{RT \ln(10)} \tag{18}$$

The relative screening is especially useful in cases where the solute is solid and the experimental free energy of fusion is unknown.

## 2.2.4 Vapor pressure and Henry's law

The saturation vapor pressure ($P_{\mathrm{sat}}$) of a pure compound ($i$) is estimated from the molar free energy of the compound in the liquid phase ($G_i^{(l)}$) and the gas phase ($G_i^{(g)}$):

$$P_{\mathrm{sat},i} = e^{-\frac{G_i^{(l)} - G_i^{(g)}}{RT}} \cdot 10^5 \,\mathrm{Pa} \tag{19}$$

COSMO*therm* calculates the infinite dilution Henry's law volatility ($H_{\mathrm{vol}}^{\infty}$, in pressure units) as a product of the pure solute saturation vapor pressure and the activity coefficient of the solute in the infinite dilution state ($\gamma_i^{\mathrm{I}}(x_i \to 0)$):

$$H_{\mathrm{vol},i}^{\infty} = P_{\mathrm{sat},i} \cdot \gamma_i^{\mathrm{I}}(x_i \to 0) \tag{20}$$

This formula is based on the assumption, that the solubility of compound $i$ in the solvent is small, allowing for the use of the zeroth order solubility approximation ($x_{\mathrm{SOL},i}^{(0)} \cong 1/\gamma_i^{\mathrm{I}}(x_i \to 0)$). Note that $\gamma_i^{\mathrm{I}}(x_i \to 0)$ is evaluated at infinite dilution, but with respect to the pure component reference state.

Using the density and molar mass of the pure solvent, Henry's law volatilities in units of pressure can be converted to Henry's law solubilities ($H_{\mathrm{sol}}^{\infty}$, in units of $\mathrm{mol\,m^{-3}\,Pa^{-1}}$):

$$H_{\mathrm{sol},i}^{\infty} = \frac{\rho}{M_{\mathrm{solvent}} \cdot H_{\mathrm{vol},i}^{\infty}} \tag{21}$$

The solvent density and molar mass are equal to the corresponding values for the solution under the assumption of infinite dilution. Densities (in $\mathrm{g\,cm^{-3}}$) of aqueous $(NH_4)_2SO_4$ and $NH_4HSO_4$ solvents in the conversion of Henry's law volatility into Henry's law solubility are calculated using the experimental polynomial fit by Tang and Munkelwitz (1994):

$$\rho = 0.9971 + \sum_{i=1}^{3} A_i (wt\%)^i \tag{22}$$

For ammonium sulfate, $A_1 = 5.92 \cdot 10^{-3}$, $A_2 = -5.036 \cdot 10^{-6}$ and $A_3 = 1.024 \cdot 10^{-8}$, and for ammonium bisulfate, $A_1 = 5.87 \cdot 10^{-3}$, $A_2 = -1.89 \cdot 10^{-6}$ and $A_3 = 1.763 \cdot 10^{-7}$.

In addition, we calculate an alternative LLE-based Henry's law solubility using the molar concentration of the solute ($c_i^{\mathrm{HA}}$) obtained from the LLE solubility calculation. This gives an estimate of the Henry's law solubility in a non-dilute solution:

$$H_{\mathrm{sol},i}^{\mathrm{LLE}} = \frac{c_i^{\mathrm{HA}}}{P_{\mathrm{sat},i}} \tag{23}$$

This definition also allows for the calculation of the effective Henry's law solubility, where the dissociation of the solute is included in the total molar concentration:

$$H_{\mathrm{sol},i}^{\mathrm{eff}} = \frac{c_i^{\mathrm{HA}} + c_i^{\mathrm{A}^-}}{P_{\mathrm{sat},i}} \tag{24}$$

# 3  Results and discussion

## 3.1  Solubility in pure water

Solubilities of organics in pure water and of water in the organic-rich phase were calculated using COSMO*therm* as the respective mole fractions at the liquid–liquid equilibrium of OS–water mixtures. Results are shown in Fig. 3.

The LLE was not found for isoprene derived organosulfates, IEPOX isomers or methyl bisulfate, indicating that these compounds are fully miscible with pure water at 298.15 K. We therefore also calculated the pure water solubilities relative to the organosulfate solubility in a 0.09 mole fraction salt solution, by solving the LLE of ternary systems where the solvent contains 0.09 mole fraction of either ammonium sulfate (AS, $(NH_4)_2SO_4$) or ammonium bisulfate (ABS, $NH_4HSO_4$). Solubility calculations for ternary systems are described in more detail in Sect. 3.2. This is done to get a quantitative estimate of the relative solubilities of the compounds which are fully soluble in pure water. The 0.09 mole fraction is below solubility limit of both $(NH_4)_2SO_4$ ($x_{SOL,AS} = 0.094$) and $NH_4HSO_4$ ($x_{SOL,ABS} = 0.33$) in water at 298.15 K (Tang and Munkelwitz, 1994). The specific inorganic salt mole fraction was chosen to be as high as possible while within the aqueous solubility limit of the salt to ensure that the organic compounds are typically not fully miscible with the salt solution. Results are shown in Fig. 3 together with corresponding binary organic solubilities. Compared to the binary LLE solubility, the aqueous solubility calculated as a relative solubility for monoterpene derived organosulfates is on average 3.1 times higher (1.8-5.5) using $(NH_4)_2SO_4$ solutions as reference, and 2.2 times higher (1.7-2.9) using $NH_4HSO_4$ solutions. The LLE was not found in the ternary systems containing IEPOX and 0.09 mole fraction of $NH_4HSO_4$.

Based on LLE calculations, the monoterpene derived organosulfates are less soluble in the ammonium sulfate and bisulfate solutions than in pure water, which means that the ammonium salts have a salting-out effect on the OS. From the solubilities calculated as relative solubility compared to $(NH_4)_2SO_4$ and $NH_4HSO_4$ solutions, we can see that the relative solubility calculation in COSMO*therm* overestimates the salting-out effect of both ammonium salts compared to the more accurate LLE calculation. In addition, the salting-out effect of $(NH_4)_2SO_4$ is overestimated more than that of $NH_4HSO_4$. The relative solubility calculation uses the zeroth order solubility approximation, which means that the estimate is less accurate when the absolute solubility of the solute is high. The largest difference between the aqueous LLE and the solubility calculated relative to the ternary LLE are seen for the OS with the higher absolute solubilities.

We see in Fig. 3 that $\alpha$-pinene-OS-1 has the lowest solubility of all the organosulfates. There are only minor structural differences between $\alpha$-pinene-OS-1 and $\alpha$-pinene-OS-2, but this still leads to a factor of 3.6 difference in the calculated solubility. All the $\beta$-pinene and limonene organosulfates, with the same functional groups as $\alpha$-pinene-OS-1 and $\alpha$-pinene-OS-2, have solubilities between those of $\alpha$-pinene-OS-1 and $\alpha$-pinene-OS-2. These results show that even minor differences in the molecular structure, such as placement of functional groups, can have a large impact on the solubility of organosulfates.

The most soluble monoterpene derived organosulfates are $\alpha$-pinene-OS-5 and $\alpha$-pinene-OS-6, that each have both a carboxylic acid group and a carbonyl group. $\alpha$-Pinene-OS-4 has a flexible carbon backbone and three carbonyl functionalities, however it still has a relatively low solubility compared to the other $\alpha$-pinene-OS. The effect of the different types of oxygen containing functional groups on the solubilities is caused by their ability to form intermolecular hydrogen bonds with the sol-

vent water. This explains the lower solubility of $\alpha$-pinene-OS-4, which has mainly hydrogen bond accepting carbonyl groups, compared to $\alpha$-pinene-OS-3, -5 and -6, which contain hydroxy groups that can act as both H-bond acceptors and donors.

We calculated acid constants ($pK_a$) for all organosulfates to capture the effect of dissociation of the neutral molecules in water. Estimated $pK_a$ values of the organosulfates are between -4.57 and -2.37, indicating that all of the organosulfates are strong acids that likely will be strongly dissociated in water. For comparison, we estimated the first $pK_a$ of sulfuric acid with COSMO*therm* to be -3.51. The organosulfates are therefore estimated to be of equivalent strength or even stronger acids than $H_2SO_4$, and thus for all practical purposes fully dissociate in near-neutral solutions and even solutions at most atmospherically relevant pH. The $pK_a$ values for all organosulfates and sulfuric acid are shown in Table S1 of the Supplement.

Dissociation corrected solubilities were calculated from Eq. (11) using the LLE solubilities in pure water and $pK_a$ estimated with COSMO*therm*. Molar liquid volumes of the pure organic compounds used to calculate densities of organic-water solutions for Eq. (12) are shown in Table S7 of the Supplement. For all organic compounds, dissociation corrected solubilities correspond to mole fractions higher than 1. This unphysical result is likely caused by inability of Eq. (11) to accurately capture solution behavior of very strongly acidic compounds. This equation is only used to calculate dissociation corrected solubilities and has no effect on other property calculations. The dissociation of strong acids is expected to be high in solutions with higher pH than the $pK_a$ of the solute (Clayden et al., 2001), such as is the case here.

Since the organosulfates are strongly dissociating in water, we also calculated the aqueous solubilities of their sodium salts (NaOS). For these sodium organosulfate salts, we here used the heat capacity of fusion estimate ($\Delta C_{p,fus} = \Delta H_{fus}/T_{melt}$) with melting point of 478.15 K (Rumble, 2018) and heat of fusion of $50\,\mathrm{kJ\,mol^{-1}}$ (Shinoda et al., 1966), respectively. Calculated solubilities of the NaOS salts are shown in Fig. 3 and Table S1 of the Supplement. For systems where a solid–liquid equilibrium was found, solubility of the organosulfate sodium salt is around 0.065 mole fraction.

## 3.2 Solubility in aqueous ammonium sulfate and bisulfate solutions

Solubilities of both OS and NaOS were calculated by solving the LLE or the SLE, respectively, in aqueous solvents containing 0.09 mole fraction of either ammonium sulfate or ammonium bisulfate. Solubility values for the OS and NaOS in these solvents, and of the aqueous ammonium sulfate and bisulfate salt solutions in the OS phase, are given in Table S2 of the Supplement.

Organic solubilities in aqueous inorganic solutions ranging from pure water to 0.09 mole fraction of inorganic salt were calculated using relative screening. These relative solubilities were then scaled using the absolute solubility values of the 0.09 mole fraction binary solvents to obtain the final relative solubilities of the OS and NaOS with respect to each binary system at the different inorganic salt mole fractions. The procedure is described in detail in Sect. S2 of the Supplement. Relative solubilities are shown in Fig. 4 (OS in $(NH_4)_2SO_4$), Fig. 5 (NaOS in $(NH_4)_2SO_4$), and Fig. S7 (OS in $NH_4HSO_4$) and Fig. S8 (NaOS in $NH_4HSO_4$) of the Supplement.

At low ($<10^{-3}$) $(NH_4)_2SO_4$ mole fractions, the molecular organosulfates are salting in, meaning that the presence of the inorganic salt enhances the total amount of the organosulfate soluble in the aqueous phase. At higher inorganic salt mole fractions the organosulfates are salting out. All IEPOX isomers and NaOS salts are salting out in the presence of co-solvated $(NH_4)_2SO_4$ across the whole concentration range. At 0.09 mole fraction of $(NH_4)_2SO_4$, the organic compounds can be grouped into three

categories based on their relative solubilities: methyl bisulfate with the highest relative solubility, isoprene derived organosulfates and IEPOX in the middle, and all monoterpene derived organosulfates with the lowest relative solubilities with respect to
370 the pure aqueous solubility.

All of the organic compounds are salting out in ternary aqueous solutions with $NH_4HSO_4$ (see Figs S7 and S8 of the Supplement) but the salting-out effect of $NH_4HSO_4$ on the organic compounds is weaker than that of $(NH_4)_2SO_4$. This is due to the stronger salting interactions of the doubly charged sulfate ion compared to the singly charged bisulfate ion.

As was mentioned above, the salting out of organosulfates from 0.09 mole fraction $(NH_4)_2SO_4$ solution is overestimated
by a factor of 3.1 using the relative solubility calculation compared to the LLE calculation. Wang et al. (2014) found that COSMO*therm* overestimates the salting-out effect of $(NH_4)_2SO_4$ on average by a factor of 3 compared to experiments. They described the salting behavior using Setschenow constants calculated from COSMO*therm* (release 14) estimated partition coefficients, which are comparable to relative solubilities. We used COSMO*therm*19 estimated relative solubilities to calculate corresponding Setschenow constants for the compounds used by Wang et al. (2014) that are in COSMO*base*17 and the same 5%
$(NH_4)_2SO_4$ solution (w/v, corresponding to $x_{AS} = 0.007$) with solvent densities by Tang and Munkelwitz (1994). We found that COSMO*therm*19 overestimates the experimental Setschenow constant of these compounds in 5% $(NH_4)_2SO_4$ solution on average by a factor of 1.5 (see Fig. S11 of the Supplement), which is an improvement to the factor of 3 of COSMO*therm*14. The overestimation might be decreased by calculating LLE solubilities as opposed to relative solubilities that use the zeroth order solubility approximation. However, finding the LLE of multiple systems is computationally infeasible and not certain to
improve the results.

Liquid–liquid phase separation (LLPS) has been detected in several aerosol experiments (Song et al., 2012; Rastak et al., 2017; Song et al., 2018; Ham et al., 2019). For example, Song et al. (2012) observed LLPS for ammonium sulfate aerosol containing organic compounds with O:C below 0.8, whereas no LLPS was seen with O:C above 0.8, depending on the functional groups. In these experiments, organic compounds contained hydroxy, carbonyl and carboxylic acid groups (Song et al., 2012).
In binary aerosol systems containing water and organic compounds (without inorganic salt), Song et al. (2018) observed LLPS for O:C below 0.44 or 0.58 in systems with one or two different organic compounds, respectively. The compounds in this study contain ester, ether and hydroxy functional groups (Song et al., 2018). With O:C ratios of the monoterpene and isoprene derived organosulfates in the ranges 0.5–0.7 and 1.2–1.4, respectively, these results are consistent with the present work. On the other hand, in experiments of OH oxidized $\alpha$-pinene and water system (Ham et al., 2019) only a single organic-rich phase was
observed, whereas LLPS was seen between water and ozone oxidized $\alpha$-pinene products (Ham et al., 2019) or OH oxidized isoprene products (Rastak et al., 2017). There are small differences in the partial charges of the oxygen atoms associated to a sulfate group compared to carboxylic acid group (see Section S3 of the Supplement for a comparison of the $\sigma$-potentials) that may influence the O:C ratio of organosulfates required for LLPS. From our results we can also see that other structural factors further affect the thermodynamic properties, in addition to the O:C ratio or the types of functional groups.

## 3.3 Activity

Activities were calculated for organosulfates and IEPOX isomers and water in binary aqueous mixtures with different organic:water molar ratios (see Table S3 of the Supplement). Figure 6 shows, as examples, the binary mixing diagrams similar to that presented by Prisle et al. (2010) of water and a) $\alpha$-pinene-OS-5, b) $\beta$-pinene-OS-1, c) limonene-OS-1, d) isoprene-OS-2 and e) $\delta_1$-IEPOX. Diagonal dashed lines illustrate the ideal mole fraction based activities ($a_i = x_i$) with respect to a pure compound ($i = $ OS, water) reference state. Since the solubility of the organic in water is much smaller than the solubility of water in the organic, the mixing diagrams for monoterpene derived organosulfates (Fig. 6a-c) are divided into two sections (note the different scales of the two phase regions): the aqueous phase (in the left panel) and the organic phase (in the right panel). In between is a composition range corresponding to the miscibility gap. From Fig. 6a-c we see how the calculated water and organosulfate activities fulfill the liquid phase equilibrium condition of Eq. (9) at the solubility limit.

Activities for the monoterpene derived organosulfates display three different types of behavior. The most common is exemplified in Fig. 6a, where in the organic-rich phase, the organosulfate activity is lower than the mole fraction of the organic ($a_{OS} \leq x_{OS}$). A low activity indicates that the organosulfate is more stable in the organic-rich phase than in the ideal pure organosulfate. The water activity is below the ideal activity ($a_w < x_w$) at low mole fractions of water and above the ideal activity ($a_w > x_w$) at higher water mole fractions, in the organic-rich phase. The organic activity at the solubility limit is low ($a_{OS} < 0.28$ when $x_{OS} = x_{SOL}$) compared to the other monoterpene derived organosulfates. Similar behavior is seen in $\alpha$-pinene-OS-3, $\alpha$-pinene-OS-4, $\alpha$-pinene-OS-6, $\beta$-pinene-2 and limonene-OS-4. A comparison between the activities of $\alpha$-pinene-OS-5 and $H_2SO_4$ calculated using COSMO*therm*, and literature values of $H_2SO_4$ activities, are shown in Fig. S9 of the Supplement.

The opposite is seen in $\alpha$-pinene-OS-1, $\beta$-pinene-OS-1 (Fig. 6b) and limonene-OS-3, where the activity of the organosulfate in the organic-rich phase is very close to or above the ideal activity. In addition, the activity at the solubility limit (both the solubility of the water and the organic) for these compounds is above 0.36. The third behavior type seen in Fig. 6c is in between the first two cases, where the water activity follows the ideal activity in small mole fractions of water. Here the organic activity at the solubility limit is around 0.3. The other compounds in this group are $\alpha$-pinene-OS-2 and limonene-OS-2.

Since the isoprene derived organosulfates, IEPOX isomers and methyl bisulfate are fully miscible with pure water, liquid–liquid phase separation was not observed for these systems. The mixing diagrams for all isoprene derived organosulfates and methyl bisulfate are similar to the one shown in Fig. 6d. Calculated activities for all IEPOX isomers are close to the ideal activities at all mixing states (Fig. 6e).

Figures 7a-e show mixing diagrams for the same organic compounds as Fig. 6a-e but now with a solvent that is 0.09 mole fraction binary aqueous solution of $NH_4HSO_4$, instead of pure water. Here, COSMO*therm* predicts LLPS also for systems containing the isoprene derived organosulfates (Fig. 7d). Again, activities for the organosulfates are higher than their mole fractions in the water-rich phase. Here we can also see that the predicted activity of water in the binary solvent is 0.78. The corresponding activity *coefficients* $\gamma_i = a_i/x_i$ for the organic compounds and water in each system of Fig. 7 are tabulated in Table S4 of the Supplement.

The calculated activity of each organic compound in the aqueous phase is higher in the ternary OS+aqueous ammonium
bisulfate systems, compared to the binary OS+water systems. This means that the inorganic salt decreases the stability of the
organosulfate in the aqueous phase. At the same time, the stability of the organosulfate in the organic-rich phase also decreases
in the presence of the inorganic salt.

Similar mixing diagrams for 0.09 mole fraction aqueous ammonium sulfate solvent are shown in Fig. S10 and tabulated
values in Table S5 of the Supplement. In ammonium sulfate solutions, COSMO*therm* predicts a water activity of 1.14 in the
aqueous solvent–rich phase, indicating that according to COSMO*therm*, the 0.09 mole fraction aqueous solution of ammonium
sulfate is unstable. This discrepancy with the experimental solubility of $x_{\mathrm{SOL,AS}} = 0.094$ (Tang and Munkelwitz, 1994) is
possibly caused by inadequate representation of the solvation of ionic liquids in COSMO-RS theory (Toure et al., 2014).

### 3.4   Saturation vapor pressure

We calculated saturation vapor pressures for the neutral organic compounds at 298.15 K (Table 1). Comparing the studied
organosulfate compounds based on their functional groups, those containing carboxylic acid groups, i.e., $\alpha$-pinene-OS-5 and
$\alpha$-pinene-OS-6, have the lowest saturation vapor pressures. $\alpha$-Pinene-OS-4, also having O:C ratio of 0.7, has an order of
magnitude higher saturation vapor pressure indicating that two carbonyl groups are less effective at lowering the vapor pressure
than a single carboxylic acid group. In addition, $\alpha$-pinene-OS-3 (one carbonyl and one hydroxy group) has a lower saturation
vapor pressure than $\alpha$-pinene-OS-4 with one more oxygen atom.
The saturation vapor pressure of sulfuric acid (extrapolated from experimental data using ab initio data) is $2.10 \cdot 10^{-3}$ Pa at
298.15 K (Ayers et al., 1980; Kulmala and Laaksonen, 1990; Noppel et al., 2002), while COSMO*therm* estimates the vapor
pressure of the pure sulfuric acid to be $7.21 \cdot 10^{-2}$ Pa (about 34 times higher). Due to the previously demonstrated systematic
over-estimation of absolute saturation vapor pressures by COSMO*therm* (Kurtén et al., 2016), we show both absolute vapor
pressures and the vapor pressures relative to the estimated sulfuric acid saturation vapor pressure in Table 1. The saturation
vapor pressures of monoterpene and isoprene derived organosulfates are all 4 to 8 orders of magnitude lower than that of
sulfuric acid. On the other hand, the saturation vapor pressures of IEPOX isomers and methyl bisulfate are higher than for
sulfuric acid.

Compared to previously calculated saturation vapor pressures for $\alpha$-pinene autoxidation products using COSMO*therm*, the
organosulfates studied here are significantly less volatile (Kurtén et al., 2016). It should be noted, however, that in the study of
Kurtén et al. (2016), the number of intramolecular H-bonds was not limited in the COSMO*therm* calculations, which likely led
to higher saturation vapor pressure estimates (Kurtén et al., 2018). Furthermore, as we have seen here, the acidic organosulfates
readily dissociate in the particle phase, forming ionic species, which will effectively suppress their partitioning to the gas phase.

$\delta_1$-IEPOX has a higher saturation vapor pressure than the other IEPOX isomers. This can be understood from a structural
point of view, as the lowest energy conformer (highest weight in the COSMO*therm* calculations) of $\delta_1$-IEPOX seems to have
two intramolecular H-bonds. COSMO*therm* does not count either of these as full or partial intramolecular hydrogen bonds in
the condensed phase. However, the gas-phase free energy ($G^{(g)}$) of the conformer is lower than for the other IEPOX conformers,

leading to about 5 kJ mol$^{-1}$ difference in the energy difference between the condensed and gas phase of $\delta_1$-IEPOX and $\delta_4$-IEPOX. This in turn leads to a relatively higher saturation vapor pressure (Eq. (19)) compared to the other IEPOX isomers.

## 3.5 Henry's law solubility

The activity coefficients at infinite dilution in water, free energies of solvation and Henry's law solubilities in pure water calculated using the different methods (explained in Sect. 2.2.4) at 298.15 K are given in Table S6 of the Supplement. Among the studied organics, Henry's law solubility is the highest for monoterpene and isoprene derived organosulfates containing the highest number of oxygen atoms and the lowest for methyl bisulfate and the IEPOX isomers. The infinite dilution Henry's law solubilities ($H_{\mathrm{sol}}^{\infty}$) were calculated by COSMO*therm* using Eq. (21). We also calculated LLE based Henry's law solubilities

($H_{\mathrm{sol}}^{\mathrm{LLE}}$) using Eq. (23) with the pure water solubilities of the organic compounds. A comparison between the infinite dilution and the LLE based Henry solubilities is shown in Fig. 8. The LLE based Henry's law solubility for monoterpene derived OS+water is on average 4.4 times lower than the corresponding infinite dilution Henry's law solubility. Henry's law solubility is the equilibrium ratio between the abundance in the gas phase and in the aqueous phase for a dilute solution. For the fully miscible compounds, and including the dissociation correction, the solution is no longer dilute. We therefore did not calculate

the LLE based Henry's law solubility of the isoprene derived compounds and methyl bisulfate, which are fully miscible with pure water at 298.15 K.

Additionally, we calculated the infinite dilution Henry's law solubilities of all compounds in two organic solvents, hexanoic and *cis*-pinonic acids (see Fig. 8 and Table S6 of the Supplement). The densities of these organic acids ($\rho_{\mathrm{hexanoic}} = 0.9400$ g cm$^{-3}$ and $\rho_{cis-\mathrm{pinonic}} = 1.0739$ g cm$^{-3}$) were estimated using COSMO*therm*. The Henry's law solubilities of the monoter-

pene derived organosulfates are the lowest in water and the highest in *cis*-pinonic acid. The isoprene derived compounds (OS and IEPOX) are all less soluble in hexanoic acid than in water. The more oxygenated isoprene-OS-3 and -4 are also less soluble in *cis*-pinonic acid than in water, opposite to the less oxygenated isoprene-OS-1 and -2, which are the most soluble in *cis*-pinonic acid. The epoxydiols are least soluble in hexanoic acid and the most soluble in water. This means that the phase separation behavior of OS from different precursors will be different in multi-phase atmospheric aerosol, leading to different

OS aerosol phase state depending on the predominant precursor.

Figure 9 shows the infinite dilution Henry's law solubilities for the organic compounds in the aqueous mixtures with different mole fractions of ammonium sulfate (left panel) and ammonium bisulfate (right panel). The decrease in Henry's law solubility is steeper with the increase of ammonium sulfate than of ammonium bisulfate. This is due to the stronger salting-out effect on the organic of ammonium sulfate than of ammonium bisulfate, seen also in the relative solubility calculations. In the case

of both inorganic salts, all of the hydroxy sulfates, i.e., $\alpha$-pinene-OS-1 and -2, and all $\beta$-pinene and limonene isomers, have similar Henry's law solubilities and trends as a function of salt mole fractions. In ammonium sulfate solutions, the Henry's law solubility of isoprene derived organosulfates decreases more slowly with the increase in ammonium salt concentration than the solubility of monoterpene derived organosulfates.

COSMO*therm* estimated Henry's law solubility has previously been reported for isoprene derived 2-methyltetrol (D'Ambro

et al., 2019), which is similar to isoprene-OS-3 and -4 with the difference that the sulfate group is replaced by a hydroxy group.

We calculated the Henry's law solubility of the 2-methyltetrol in water using COSMO*therm*19 and found that the compounds containing a sulfate group (isoprene-OS-3 and -4) have Henry's law solubilities that are 4 orders of magnitude higher than the compound containing only hydroxy groups (2-methyltetrol). The higher Henry's law solubility of the organosulfate, compared to the 2-methyltetrol, is caused by 5 orders of magnitude lower saturation vapor pressure and an order of magnitude higher activity coefficient at the infinite dilution of the solute. Similar differences are seen between the IEPOX isomers and isoprene-OS-1 and -2, although the functional groups in isoprene-OS-1 and -2 (hydroxy and carbonyl) are different than those in IEPOX (hydroxy and epoxy). This means that the presence of sulfate in SOA and the formation of organosulfate compounds enhances SOA formation, since organosulfates are less likely to evaporate than non-sulfate organics.

## 4   Conclusions

We have used COSMO*therm* to evaluate thermochemical properties ($pK_a$, solubility, activity, Henry's law solubility and saturation vapor pressure) of organosulfates derived from isoprene, $\alpha$-pinene, $\beta$-pinene and limonene. These properties are key to governing the phase-state behavior and stability of organosulfates as components of atmospheric SOA.

Interactions with atmospheric water is a critical process determining the growth of SOA and in turn any size-dependent effects, such as heterogeneous chemistry mediated by available surface area and both direct and indirect climate effects of aerosols. The studied organosulfates have several polar functional groups and in many cases amphiphilic structures. Overall, the organosulfates display both favorable ($a_{OS} < x_{OS}$) and unfavorable ($a_{OS} > x_{OS}$) interactions with water in the condensed phase. Both behaviors are seen for the same compound in different regions of the mixing diagram. In water+monoterpene derived organosulfate mixtures, COSMO*therm* predicts phase-separation into organic-rich and water-rich phases. Particles with LLPS have already been detected in field samples and generated in numerous laboratory experiments when mixing inorganic sulfate salts and organic compounds (e.g. carboxylic acids and electrolytes, organosulfates from VOC oxidation) (Wu et al., 2018; Bondy et al., 2018). When a miscibility gap exists, water uptake to the organic-rich aerosol phase, as well as organosulfate formation in the aqueous aerosol, is not a continuous function of relative humidity or organosulfate precursor availability.

In the particular case of cloud condensation nuclei (CCN) activation, elevated water activities in a water-rich phase due to the presence of organosulfate solute will suppress water uptake from a decreased Raoult effect ($a_w > x_w$) and decrease SOA CCN activity. However, interactions may not be constant across the phase diagram. Variations between organic-rich and water-rich phases, as well as between the organosulfates, can contribute to explain the variation in limonene-derived OS hygroscopicity parameter between sub- and supersaturated conditions observed by Hansen et al. (2015). They also found a non-linear composition dependence of the CCN activity of mixed OS-AS aerosols and connected the inability of their Köhler model to capture this trend with non-ideal behavior of the droplet solutions (Hansen et al., 2015). The COSMO*therm* estimated activities can be used in Köhler calculations to model the non-ideality of aqueous droplet solutions. For instance, hygroscopic growth factor (calculated as a ratio between wet and dry particle diameter) is higher for particles with lower water activity than for particles with higher water activity. From our calculation we can see that for some of the organosulfates, water activity in the organic phase is above ideality ($a_w > x_w$), meaning a lower water uptake compared to the organosulfates with water

activities below ideality ($a_w < x_w$). Additionally, a miscibility gap means that the aerosol system has inaccessible mixing states. Therefore, not all conditions, including the CCN activation threshold, may be reached in a continuous fashion during cloud processing but could instead be short-circuited by aerosol LLPS.

Our calculations predict limited organosulfate solubility in pure water, and even lower solubility in the aqueous solutions of ammonium sulfate and ammonium bisulfate. Solubility is however strongly enhanced by formation of the corresponding organosulfate anionic species, in aqueous environments which are not very strongly acidic. Previous experimental, modelling and computational studies (Wang et al., 2014, 2016) have shown that ammonium sulfate has a salting out effect on organic compounds. This is seen in our calculations for the IEPOX isomers, whereas a weak (at most 3.5%) salting-in effect on the organosulfates is predicted at low concentrations of ammonium sulfate. COSMO*therm* has previously been shown to overestimate the salting-out effect of ammonium sulfate on diverse organic compounds (Wang et al., 2014; Toivola et al., 2017). Based on this, it is possible that the salting-in of organosulfates may be underestimated in our present calculations. Presence of additional inorganic salt in the aerosol where organosulfate is formed may therefore both enhance or decrease the SOA phase stability of the organosulfate, depending on the organic:inorganic sulfate mixing ratio and relative humidity.

Calculated saturation vapor pressures are lower for organosulfates than isoprene derived dihydroxy dihydroperoxides and dihydroperoxy hydroxy aldehydes (Kurtén et al., 2018) and $\alpha$-pinene derived oxidized compounds (Kurtén et al., 2016). Based on this, organosulfates are more stable in the condensed phase than non-sulfate organic compounds. In addition, the saturation vapor pressure of $H_2SO_4$ is higher than all of the organosulfates. Due to the low p$K_a$ of all organosulfates (and $H_2SO_4$), if the aerosol contains molecules or ions capable of acting as bases, the effective vapor pressure (equilibrium vapor pressure) of OS SOA will be many orders of magnitude lower than the saturation vapor pressures. Overall, organosulfates are thus unlikely to evaporate from an aerosol in which they are formed. This means that the formation of organosulfates, and in particular the formation of their salts, can contribute significantly to increasing the SOA mass in regions with high sulfate aerosol content. Not only will OS add to SOA, this SOA will also be stable over a wide range of conditions, including salinity and acidity.

Results of this work show that COSMO*therm* is a viable path to obtaining compound-specific thermochemical properties of atmospheric organic aerosol, which may not be available through experimental methods any time soon. We have calculated values for selected properties which are overall consistent with observations from a variety of aerosol measurements from both field and laboratory work. However, we also see that oxidized atmospheric organics from similar precursors and with similar chemical functionalities may exhibit surprisingly different compound-specific phase-state properties. In combination with the variation of these properties across a range of conditions, this thermochemical heterogeneity of atmospheric organosulfates - as of other compound classes which may display similar variation - poses a real challenge for large-scale atmospheric simulations. In particular, we note that great caution must be taken when using single compounds to represent the properties of an entire group.

*Data availability.* The research data has been deposited in a reliable public data repository (the CERN Zenodo service) and can be accessed at: https://doi.org/10.5281/zenodo.3552309 (Hyttinen et al., 2019).

*Author contributions.* NLP conceived, planned, supervised and secured funding for the project. NH performed the calculations. NH and NLP analyzed the results and wrote the paper, with contributions from the other authors. NH revised the paper with input from the other authors.

*Competing interests.* The authors declare that they have no conflict of interest.

*Acknowledgements.* We thank Doc. Theo Kurtén for helpful discussions, and Dr. Frank Eckert and Dr. Jens Reinisch for advice in the use of COSMO*therm*. This project has received funding from the European Research Council (ERC) under the European Union's Horizon 2020 research and innovation programme, Project SURFACE (Grant Agreement No. 717022). The authors also gratefully acknowledge the financial contribution from the Academy of Finland, including Grant No. 308238 and 314175. J. E. thanks the Swedish Research Council 575 Formas project number 2018-01745-COBACCA for financial support. We thank CSC - IT Center for Science, Finland, for computational resources.

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

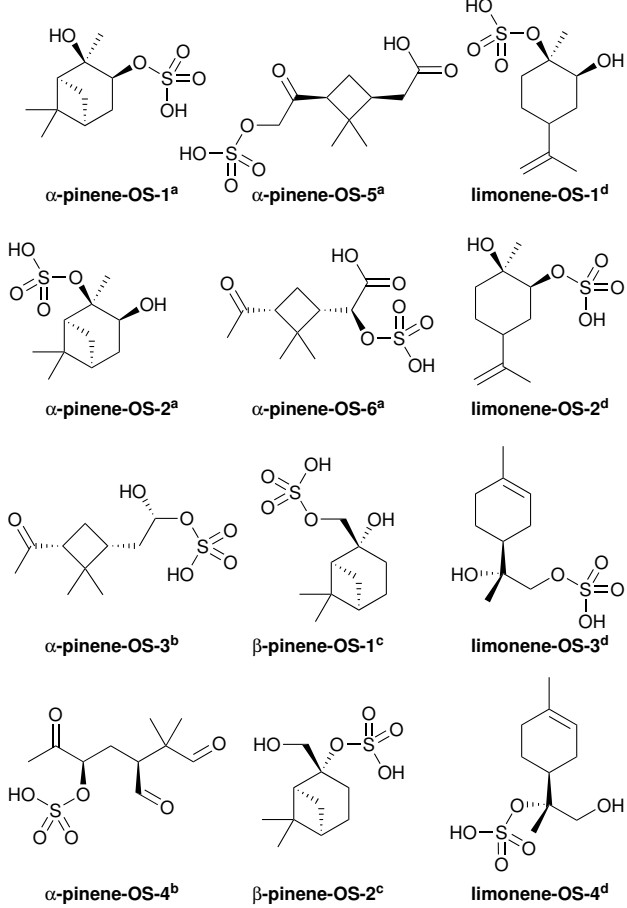

**Figure 1.** Structures of the studied monoterpene derived organosulfates, provided by [a]Surratt et al. (2008), [b]Surratt et al. (2007), [c]Iinuma et al. (2009) and [d]Hansen et al. (2015).

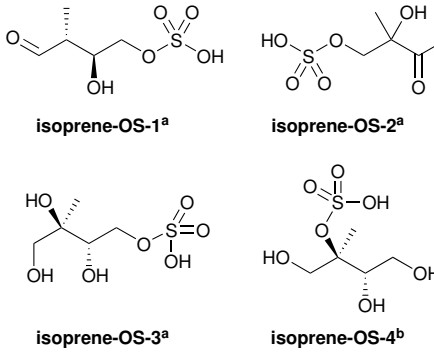

**isoprene-OS-1[a]**        **isoprene-OS-2[a]**

**isoprene-OS-3[a]**        **isoprene-OS-4[b]**

**Figure 2.** Structures of the studied isoprene derived organosulfates, provided by [a]Surratt et al. (2007) and [b]Surratt et al. (2010). Isoprene-OS-3 and isoprene-OS-4 are IEPOX derived organosulfates.

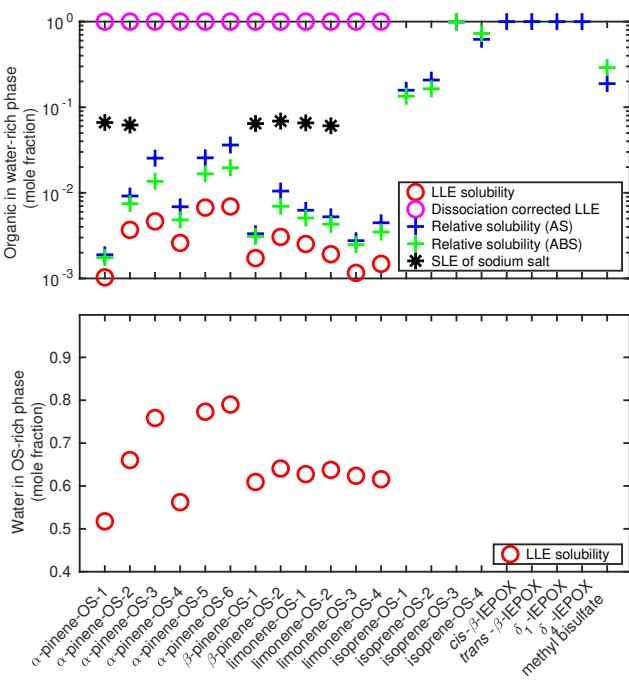

**Figure 3.** In the top panel, solubility of organosulfates and their sodium salts in pure water, and in the bottom panel, the solubility of water in the organosulfate phase ($T = 298.15$ K). Solubilities were estimated using the SLESOL method to solve the liquid–liquid (LLE) or solid–liquid (SLE) equilibrium in COSMO*therm*. LLE/SLE was not found for the systems with missing points, indicating that the solute is fully miscible with the solvent. Relative solubilities of organosulfates and IEPOX were calculated using the LLE solubility of each compound in 0.09 mole fraction of the inorganic salt (AS or ABS) as reference for the pure water solubility.

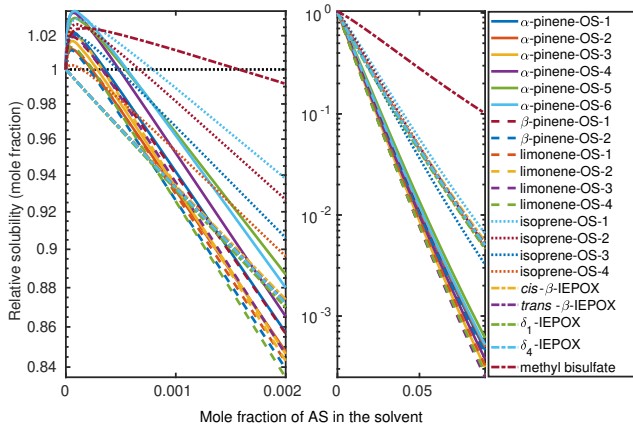

**Figure 4.** Relative solubilities of OS in $(NH_4)_2SO_4$ (aq) solutions ($T$ = 298.15 K, relative to pure water) estimated in COSMO*therm* using relative screening. The left panel shows results for the lower binary salt mole fraction range from 0 to $2 \cdot 10^{-3}$, and the right panel shows the whole range between 0 and 0.09 mole fraction of the salt. The black dotted line in the left panel shows the relative solubility = 1, equivalent to the solubility of the OS in pure water.

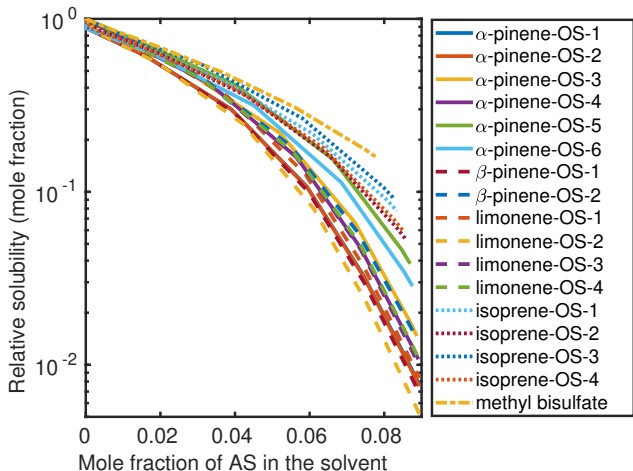

**Figure 5.** Relative solubilities of NaOS salts in $(NH_4)_2SO_4$ (aq) solutions ($T$ = 298.15 K, relative to pure water) estimated using relative screening in COSMO*therm*.

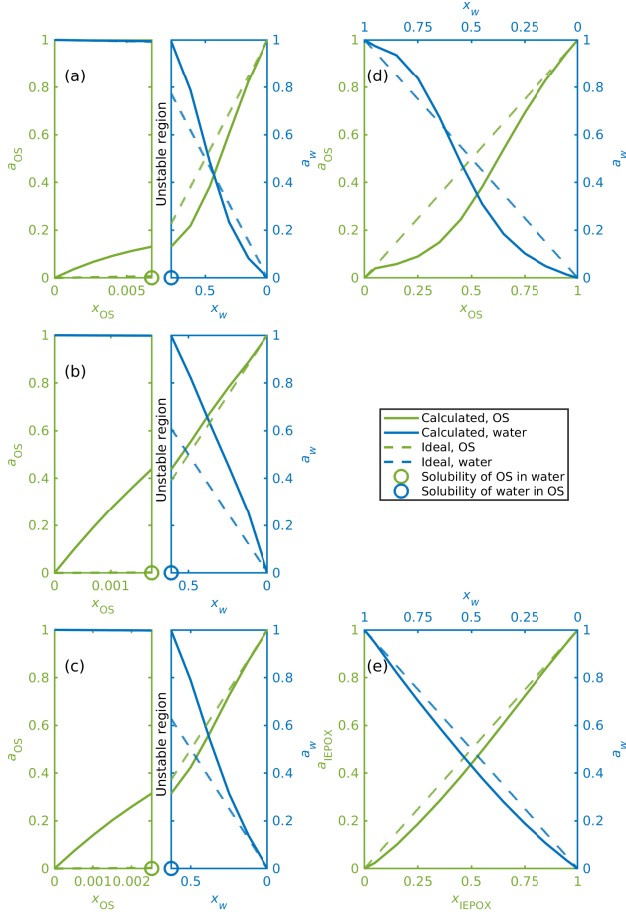

**Figure 6.** Activities of OS, IEPOX and water in binary mixtures. a) $\alpha$-pinene-OS-5, b) $\beta$-pinene-OS-1, c) limonene-OS-1, d) isoprene-OS-2, e) $\delta_1$-IEPOX. The left hand sides of panels a-c show the water-rich phase and the right hand sides the corresponding organic-rich phase.

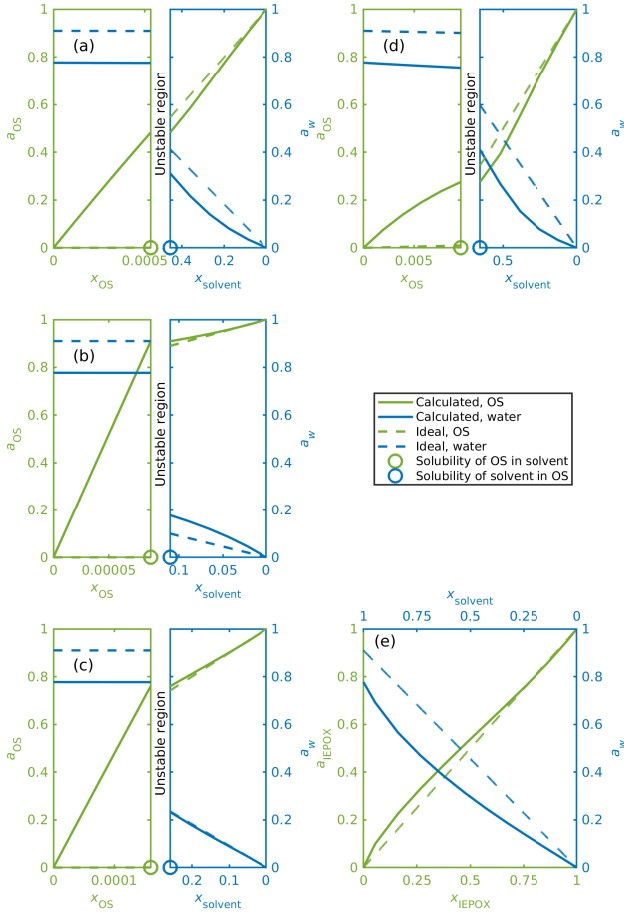

**Figure 7.** Activities for OS, IEPOX and water in ternary aqueous mixtures. The solvent is a 0.09 mole fraction ammonium bisulfate solution and the ideal water activity is equal to the mole fraction of water. a) $\alpha$-pinene-OS-5, b) $\beta$-pinene-OS-1, c) limonene-OS-1, d) isoprene-OS-2, e) $\delta_1$-IEPOX. The left hand sides of panels a-d show the solvent-rich phase and the right hand sides the organic-rich phase. The ABS:water ratio is kept constant in all calculated mixing states, which means that ammonium bisulfate and water are not individually at equilibrium at the solubility limits.

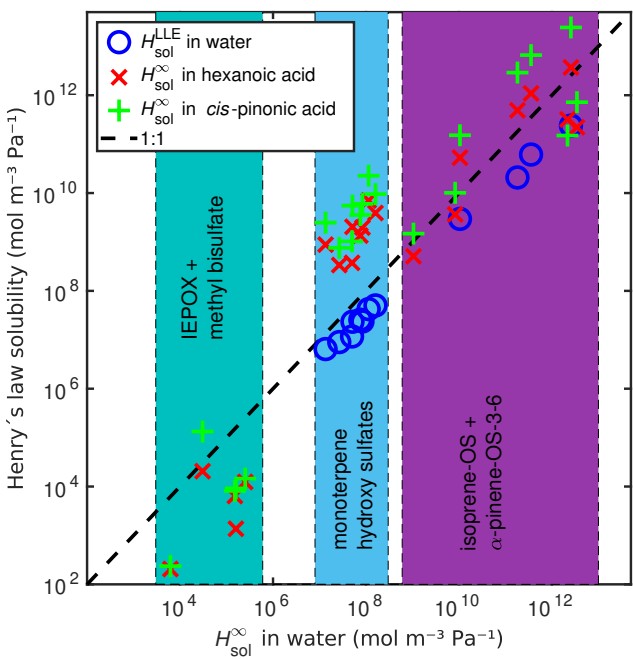

**Figure 8.** Comparison between infinite dilution Henry's law solubility ($H_{\mathrm{sol}}^{\infty}$) in water, hexanoic acid and *cis*-pinonic acid, and LLE based Henry's law solubility ($H_{\mathrm{sol}}^{\mathrm{LLE}}$) in water. The dashed line shows 1:1 ratio between $H_{\mathrm{sol}}^{\infty}$ in water and the other Henry's law solubilities.

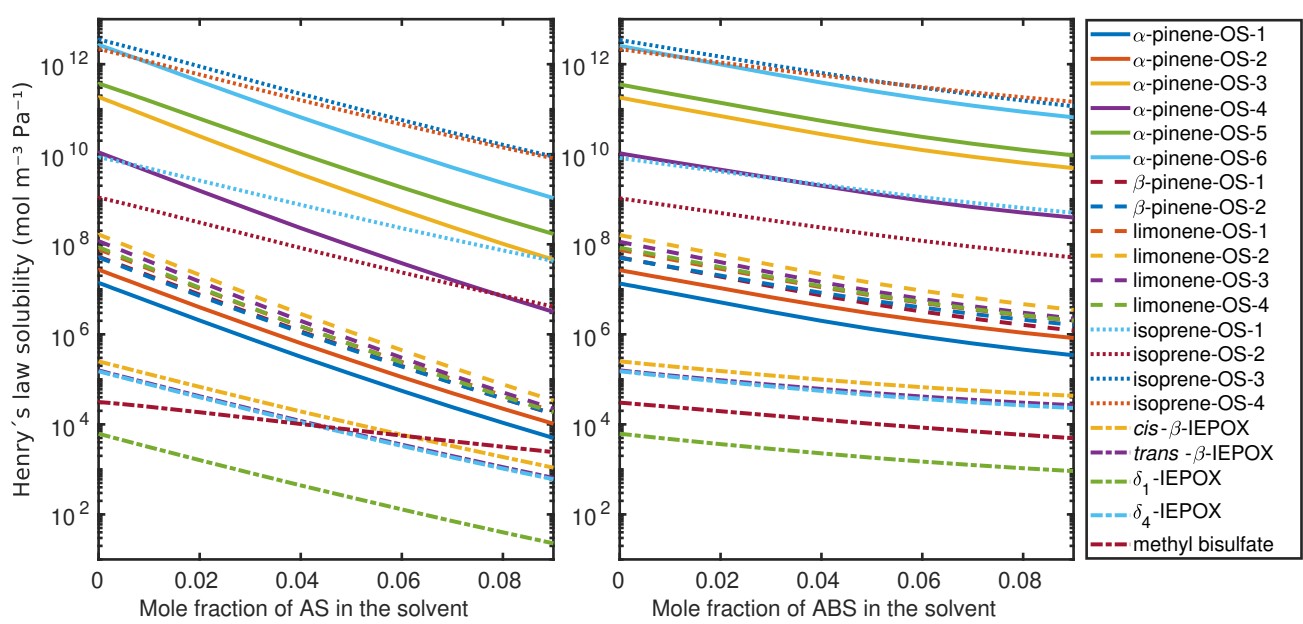

**Figure 9.** Infinite dilution Henry's laws solubilities in aqueous solutions of $(NH_4)_2SO_4$ (left panel) and $NH_4HSO_4$ (right panel), at 298.15 K.

**Table 1.** Estimated saturation vapor pressures of the pure compounds, and the ratio between the saturation vapor pressures of the organic compound and sulfuric acid

| compound | $P_{\text{sat}}$ (Pa) | $\frac{P_{\text{sat}}}{P_{\text{sat},\text{H}_2\text{SO}_4}}$ |
|---|---|---|
| $\alpha$-pinene-OS-1 | $7.96 \cdot 10^{-6}$ | $1.10 \cdot 10^{-4}$ |
| $\alpha$-pinene-OS-2 | $2.00 \cdot 10^{-5}$ | $2.78 \cdot 10^{-4}$ |
| $\alpha$-pinene-OS-3 | $1.08 \cdot 10^{-8}$ | $1.49 \cdot 10^{-7}$ |
| $\alpha$-pinene-OS-4 | $4.31 \cdot 10^{-8}$ | $5.98 \cdot 10^{-7}$ |
| $\alpha$-pinene-OS-5 | $5.19 \cdot 10^{-9}$ | $7.20 \cdot 10^{-8}$ |
| $\alpha$-pinene-OS-6 | $1.37 \cdot 10^{-9}$ | $1.90 \cdot 10^{-8}$ |
| $\beta$-pinene-OS-1 | $3.65 \cdot 10^{-6}$ | $5.07 \cdot 10^{-5}$ |
| $\beta$-pinene-OS-2 | $1.28 \cdot 10^{-5}$ | $1.78 \cdot 10^{-4}$ |
| limonene-OS-1 | $4.84 \cdot 10^{-6}$ | $6.72 \cdot 10^{-5}$ |
| limonene-OS-2 | $1.88 \cdot 10^{-6}$ | $2.61 \cdot 10^{-5}$ |
| limonene-OS-3 | $1.36 \cdot 10^{-6}$ | $1.89 \cdot 10^{-5}$ |
| limonene-OS-4 | $3.10 \cdot 10^{-6}$ | $4.31 \cdot 10^{-5}$ |
| isoprene-OS-1 | $1.68 \cdot 10^{-6}$ | $2.33 \cdot 10^{-5}$ |
| isoprene-OS-2 | $2.15 \cdot 10^{-5}$ | $2.98 \cdot 10^{-4}$ |
| isoprene-OS-3 | $2.42 \cdot 10^{-8}$ | $3.36 \cdot 10^{-7}$ |
| isoprene-OS-4 | $2.07 \cdot 10^{-8}$ | $2.87 \cdot 10^{-7}$ |
| *cis*-$\beta$-IEPOX | 0.235 | 3.26 |
| *trans*-$\beta$-IEPOX | 0.392 | 5.43 |
| $\delta_1$-IEPOX | $2.35 \cdot 10^{1}$ | $3.26 \cdot 10^{2}$ |
| $\delta_4$-IEPOX | 0.441 | 6.12 |
| methyl bisulfate | 1.04 | $1.44 \cdot 10^{1}$ |
| sulfuric acid | $7.21 \cdot 10^{-2}$ | 1 |