# Peer review of "Thermodynamic properties of isoprene and monoterpene derived organosulfates estimated with COSMOtherm"

_Atmospheric Chemistry and Physics, 2019_

## Referee Comment (RC1) · Anonymous Referee #1 · 20 Jan 2020

General Comments:

Isoprene and monoterpene organosulfates contribute significantly to atmospheric secondary aerosol formation. However, their physical chemical properties are rarely studied, due to the difficulty in isolating adequate quantity of individual chemicals and challenge in the measurement. In this paper, the authors used computation method COSMOtherm to predict thermodynamic properties for organofulfates, including solubilities, activities and saturation vapor pressures, pKa, salting out effect. The authors provide adequate information on the technical details. A major comment is that the authors should add discussion about the uncertainties of the predicted values for dif-

ferent properties. This could be either from literature that have used COSMOtherm to predict different physical-chemical properties (e.g. Henry's law constants, solubility, pKa, salting out effect, saturation vapor pressure, etc.) or from the own estimation.

Specific comments:

Line 67, the authors may consider describing COSMOtherm more here, since majority of the readers of this paper are not familiar with the program.

Line 104, in reality, OS likely have conformers with intramolecular hydrogen bonds, what is the uncertainty if the intramolecular hydrogen bonding is not considered (or only include conformers containing no intramolecular hydrogen bonds) in the calculation? In addition, is the version of COSMOtherm program used by Kurten et al. (2018) the same as in this paper, if not, are there any improvement with intramolecular hydrogen bond representation in the updated version of COSMOtherm?

Section 3.2, it was mentioned early that organosulfates dissociate significantly in pure water, did the authors consider dissociation of organosulfates when calculating solubility in ammonium sulfate or bisulfate solutions?

Line 206, it seems that the authors used hydrated sodium cations for the description of sodium cation solvation for organosulfate sodium salts. Did they use hydrated salt ions for ammonium sulfate and ammonium bisulfate? If not, why?

Line 263 and others, the authors mentioned 0.09 mole fraction salt solution in multiple places in the paper. Why is 0.09 mole fraction used? Is this the saturation concentration of a salt in water?

Line 413, Figure 4 and 9 show the influence of ammonium sulfate and ammonium bisulfate on solubility and Henry's law constants for different species. I understand that there are no experimental data available to validate the predicted values for the species of interest. However, the authors should do calculations for chemicals with experimental data available to validate the model prediction. Especially, since the previous publications (Endo et al. 2012, Wang et al., 2014, Toivola et al., 2017) on the topic of using COSMOtherm to predict salt out effect observe an overestimation of salting out effect with COSMOtherm in comparison to experimental values.

Endo, S., Pfennigsdorff, A., Goss, K-U. Salting-out effect in aqueous NaCl solutions: trends with size and polarity of solute molecules. Environ. Sci. Technol., 2012, 46, 3 1496-1503.

Wang, C., Lei, Y. D., Endo, S., Wania, F. Measuring and modeling the salting-out effect in ammonium sulfate solutions, Environ. Sci. Technol., 2014, 48, 13 238-13 245.

Toivola, M., Prisle, N. L., Elm, J., Waxman, E. M., Volkamer, R., and Kurtén, T.: Can COSMOTherm Predict a Salting in Effect? J. Phys.Chem. A, 2017, 121, 6288–6295.

The authors predicted the Henry's law constant for organosulfates in water. However, in the atmosphere there are also large volume of organic phase available. What about the Henry's law constant into organic phase?

---

## Referee Comment (RC2) · Anonymous Referee #2 · 13 Feb 2020

In this work, the authors have used the COSMOtherm program to calculate both properties of atmospherically relevant monoterpene and isoprene derived organosulfates (such as solubilities, activities and saturation vapor pressures). The new modeled results are important for us to better understand the atmospheric impacts and fates of organosulfates. The model simulations are carefully designed and run with strong justifications and assumptions. The paper is well written and is suitable for publication in ACP. I have two major suggestions:

1. It is understood that there are uncertainties for the model simulations. The authors have done a very nice work in explaining your model simulations. However, it would

be still useful for the readers to know what would be the potential uncertainties of the modeled results (e.g. what is the possible range of the model results instead of a single value?).

2. To date, it remains unclear how the sulfate group would affect the properties of the organic compounds. It would be useful if the authors could discuss how the presence or absence of sulfate group would change the properties of an organic molecules based on their model simulations.

Minor comments:

Abstract, "The estimated pKa values of all the organosulfates indicate a high degree of dissociation in water, leading in turn to high dissociation corrected solubilities." Any explanation from the model simulations?

Line 45, "The hygroscopic properties of organosulfate containing aerosol have been measured using sodium salts of alkane sulfates (Woods III et al., 2007; Estillore et al., 2016) and limonene derived organosulfates (Hansen et al., 2015). Limonene derived organosulfate was demonstrated to lower the surface tension of aqueous solutions even more effectively than atmospherically relevant strong organic acids (Hansen et al., 2015)." Could the authors compare their model results with some of these literature data?

Line 47, "In addition, Nguyen et al. (2014a, b) have seen indications of long-range transport of organosulfates, suggesting that organosulfates must have sufficiently low volatilities to remain in the aerosol-phase over a wide range of atmospheric conditions." This observation does not consistent with the model results? As stated in the abstract, "Based on the estimated saturation vapor pressures, the organosulfates of this study can all be categorized as semi- or low-volatile, with saturation vapor pressures 4 to 8 orders of magnitude lower than that of sulfuric acid". Do these modelled results support by the laboratory and field observations?

Line 80, "The non-systematic conformer generation in COSMOconf has been shown to lead to significantly different results in COSMOtherm depending on the initial geometry with molecules containing hydroxy and hydroperoxy functional groups (Kurtén et al., 2018). Based on the recommendation by Kurtén et al. (2018), we therefore used the systematic conformer sampling with the MMFF force fields in the Spartan '14 program (Wavefunction Inc., 2014)." While the organosulfates contain a sulfate group, would the recommendation made by Kurtén et al. (2018) be the best option for the organosulfates investigated in this work? More elaboration would be needed.

Line 96, "All calculations were done at 298.15 K and we assume that all of the organosulfates (OS) and the isoprene epoxydiols (IEPOX) are liquids." Please elaborate why we could assume all organosulfates investigated in the work are liquids in their pure forms at 298K.

Line 104, "Kurtén et al. (2018) found that COSMOtherm overestimates the effect of intramolecular hydrogen bonds and recommended that only conformers containing no intramolecular hydrogen bonds should be used in saturation vapor pressure calculations. We therefore omitted all conformers containing intramolecular hydrogen bonds from the calculations of OS and IEPOX." This argument is okay. However, how this assumption would affect the model results?

Line 268, "Compared to the binary LLE solubility, the solubility calculated as a relative solubility for monoterpene derived organosulfates is on average 3.1 times higher (1.8-5.5) using (NH4)2SO4 solutions as reference, and 2.2 times higher (1.7-2.9) using NH4HSO4 solutions." Could the authors elaborate why the relative solubility for monoterpene derived organosulfates is higher than ammonium sulfate and ammonium bisulfate?

Line 286, "The organosulfates are therefore estimated to be of equivalent strength or even stronger acids than H2SO4, and thus for all practical purposes fully dissociate in near-neutral solutions and even solutions at most atmospherically relevant pH." These

are important results. Any literature data to support these findings?

Line 288, "For all organic compounds, dissociation corrected solubilities correspond to mole fractions higher than 1. This unphysical result is likely caused by inability to accurately capture solution behavior of very strongly acidic compounds." How would this factor affect other modelled results?

Line 326, "With O:C ratios of the monoterpene and isoprene derived organosulfates in the ranges 0.5–0.7 and 1.2–1.4, respectively, these results are consistent with the present work. On the other hand, in experiments of OH oxidized $\alpha$-pinene and water system (Ham et al., 2019) only a single organic-rich phase was observed, whereas LLPS was seen between water and ozone oxidized $\alpha$-pinene products (Ham et al., 2019) or OH oxidized isoprene products (Rastak et al., 2017)." For organosulfates, 4 oxygen atoms are associated from the sulfate group. Would this affect how we interpret the O:C ratio?

Line 375, "We calculated saturation vapor pressures for the neutral organic compounds at 298.15 K (Table 1). Comparing the studied organosulfate compounds based on their functional groups, those containing carboxylic acid groups, i.e., $\alpha$-pinene-OS-5 and $\alpha$-pinene-OS-6, have the lowest saturation vapor pressures." How the presence of sulfate group would affect the saturation vapor pressure of the organic compounds?

Line 403, "Among the studied organics, Henry's law solubility is the highest for monoterpene and isoprene derived organosulfates containing the highest number of oxygen atoms and the lowest for methyl bisulfate and the IEPOX isomers." Also, how the presence of sulfate group would affect the Henry's law solubility of the organic compounds?

---

## Author Comment (AC1) · 25 Mar 2020

We thank the referees for their thoughtful and constructive comments which we believe have helped improve the manuscript substantially. We have revised the manuscript following the referees' suggestions. You can find answers to the referee comments (text in red) below with additions to the manuscript and supplement text (**in bold**). The added references are listed at the end.

**Anonymous Referee #1**

Isoprene and monoterpene organosulfates contribute significantly to atmospheric secondary aerosol formation. However, their physical chemical properties are rarely studied, due to the difficulty in isolating adequate quantity of individual chemicals and challenge in the measurement. In this paper, the authors used computation method COSMOtherm to predict thermodynamic properties for organosulfates, including solubilities, activities and saturation vapor pressures, pKa, salting out effect. The authors provide adequate information on the technical details. A major comment is that the authors should add discussion about the uncertainties of the predicted values for different properties. This could be either from literature that have used COSMOtherm to predict different physical-chemical properties (e.g. Henry's law constants, solubility, pKa, salting out effect, saturation vapor pressure, etc.) or from the own estimation.

**Changes in manuscript: The accuracy of COSMO*therm* p$K_a$ calculations (parametrization BP_TZVPD_FINE_C30_16.01) is 0.65 log units RMSD (Klamt et al., 2016) and experimental saturation vapor pressures can be predicted within a factor of 2 with a tendency to overpredict experimental values of carboxylic acids (Schröder et al., 2016). For citric acid, using only conformers containing no intramolecular H-bonds leads to good agreement between experiments and COSMO*therm* estimated aqueous solubility, activity coefficients, density and p$K_a$ (Hyttinen and Prisle, 2020).**

Specific comments:

Line 67, the authors may consider describing COSMOtherm more here, since majority of the readers of this paper are not familiar with the program.

**Changes in manuscript: COSMO*therm* combines quantum chemistry and statistical thermodynamics to predict condensed-phase properties of liquids as well as partitioning between condensed and gas phases. Quantum chemical calculations provide input files (cosmo-files) for COSMO*therm* and the same files can be used to estimate properties in various solutions. In addition, multiple conformers can be included in COSMO*therm* calculations to improve the description of conformer distributions in different solutions.**

Line 104, in reality, OS likely have conformers with intramolecular hydrogen bonds, what is the uncertainty if the intramolecular hydrogen bonding is not considered (or only include conformers containing no intramolecular hydrogen bonds) in the calculation?

**Changes in manuscript: In COSMO*therm* calculations, conformers are weighted according to the Boltzmann distribution based on the sum of their solvated energy and chemical potential in the solution. However, normally only conformers with lowest solvated energies are selected for COSMO*therm* calculations. If the total number of unique conformers is high, not all conformers can be included in the COSMO*therm* calculation. When only a fraction of all conformers is used in a COSMO*therm* calculation, only those containing intramolecular H-bonds are used, as they have the lowest solvated energies. However, the interaction between a compound and water is more favorable for conformers containing no intramolecular H-bonds. Therefore, in aqueous solutions, the chemical potential of conformers containing no intramolecular H-bonds is much lower than of conformers that contain multiple H-bonds (Hyttinen and Prisle, 2020). If a compound contains more unique conformers than can be included in COSMO*therm* calculations, more attention should be paid to selecting the conformers to represent the conformer distribution in the studied solutions.**

**Generally, the omission of conformers containing intramolecular H-bonds leads to lower saturation vapor pressures (Kurtén et al., 2018), higher aqueous solubility and larger deviation from ideality of activity coefficients (Hyttinen and Prisle, 2020).**

In addition, is the version of COSMOtherm program used by Kurten et al. (2018) the same as in this paper, if not, are there any improvement with intramolecular hydrogen bond representation in the updated version of COSMOtherm?

**Changes in manuscript: We tested the difference in saturation vapor pressures calculated using releases 18 and 19 (parametrizations BP_TZVPD_FINE_18 and BP_TZVPD_FINE_19, respectively) and found that differences between the two parametrizations are larger using all conformers than when only conformers containing no intramolecular H-bonds are used. Variation between estimates using different conformer sets is also smaller in release 19 than in release 18. Hyttinen and Prisle (2020) found a good agreement between experimental and COSMO*therm* (release 19) estimated solubilities and activity coefficients when conformers containing intramolecular H-bonds were omitted from the COSMO*therm* calculations.**

Section 3.2, it was mentioned early that organosulfates dissociate significantly in pure water, did the authors consider dissociation of organosulfates when calculating solubility in ammonium sulfate or bisulfate solutions?

**Changes in manuscript: The parametrization in COSMO*therm* currently enables calculation of p$K_a$ only in water, dimethylsulfoxide, acetonitrile or heptane. We are therefore not able to estimate p$K_a$ values of the organosulfates in other solvents relevant to this work, specifically aqueous ammonium sulfate and bisulfate solutions.**

Line 206, it seems that the authors used hydrated sodium cations for the description of sodium cation solvation for organosulfate sodium salts. Did they use hydrated salt ions for ammonium sulfate and ammonium bisulfate? If not, why?

**Changes in manuscript: In COSMO*therm*, small atomic metal ions have extreme screening charge densities ($\sigma < -0.025$ e Å$^{-2}$ or $\sigma > 0.025$ e Å$^{-2}$). In reality, extreme screening charge densities of ions would lead to the formation of a solvation shell, where polar solvent molecules form strong H-bonds with the ion. This is not accounted for in COSMO*therm*, which leads to unrealistic behavior of the sodium ion in water.**

**The screening charge densities of larger ions, such as ammonium, sulfate and bisulfate, are less extreme ($-0.025$ e Å$^{-2} < \sigma < 0.025$ e Å$^{-2}$, see Fig. S4 of the Supplement) and the non-hydrated ions can be used in COSMO*therm* calculations.**

**Changes in Supplement:**

[Figure]

**Figure S4: σ-profiles of the inorganic ions used in the COSMO*therm* calculations. Negative σ values (screening charge density) indicate a positive partial charge and positive σ values negative partial charge.**

Line 263 and others, the authors mentioned 0.09 mole fraction salt solution in multiple places in the paper. Why is 0.09 mole fraction used? Is this the saturation concentration of a salt in water?

**Changes in manuscript: The 0.09 mole fraction is below the solubility limit of both $(NH_4)_2SO_4$ ($x_{SOL,AS}=0.094$) and $NH_4HSO_4$ ($x_{SOL,ABS}=0.33$) in water at 298.15 K (Tang and Munkelwitz, 1994). The specific inorganic salt mole fraction was chosen to be as high as possible while within the aqueous solubility limit of the salt to ensure that the organic compounds are typically not fully miscible with the salt solution.**

Line 413, Figure 4 and 9 show the influence of ammonium sulfate and ammonium bisulfate on solubility and Henry's law constants for different species. I understand that there are no experimental data available to validate the predicted values for the species of interest. However, the authors should do calculations for chemicals with experimental data available to validate the model prediction. Especially, since the previous publications (Endo et al. 2012, Wang et al., 2014, Toivola et al., 2017) on the topic of using COSMOtherm to predict salt out effect observe an overestimation of salting out effect with COSMOtherm in comparison to experimental values.

Endo, S., Pfennigsdorff, A., Goss, K-U. Salting-out effect in aqueous NaCl solutions:trends with size and polarity of solute molecules. Environ. Sci. Technol., 2012, 46, 31496-1503.

Wang, C., Lei, Y. D., Endo, S., Wania, F. Measuring and modeling the salting-out effectin ammonium sulfate solutions, Environ. Sci. Technol., 2014, 48, 13 238-13 245.

Toivola, M., Prisle, N. L., Elm, J., Waxman, E. M., Volkamer, R., and Kurtén, T.: Can COSMOTherm Predict a Salting in Effect? J. Phys.Chem. A, 2017, 121, 6288–6295.

**Changes in manuscript: As was mentioned above, the salting out of organosulfates from 0.09 mole fraction $(NH_4)_2SO_4$ solution is overestimated by a factor of 3.1 using the relative solubility calculation compared to the LLE calculation. Wang et al. (2014) found that COSMO*therm* overestimates the salting-out effect of $(NH_4)_2SO_4$ on average by a factor of 3 compared to experiments. They described the salting behavior using Setschenow constants calculated from COSMO*therm* (release 14) estimated partition coefficients, which are comparable to relative solubilities. We used COSMO*therm*19 estimated relative solubilities to calculate corresponding Setschenow constants for the compounds used by Wang et al. (2014) that are in COSMO*base*17 and the same 5% $(NH_4)_2SO_4$ solution (w/v, corresponding to $x_{AS}$=0.007) with solvent densities by Tang and Munkelwitz (1994). We found that COSMO*therm*19 overestimates the experimental Setschenow constant of these compounds in 5% $(NH_4)_2SO_4$ solution on average by a factor of 1.5 (see Fig. S11 of the Supplement), which is an improvement to the factor of 3 obtained with COSMO*therm*14. The overestimation might be decreased by calculating LLE solubilities as opposed to relative solubilities that use the zeroth order solubility approximation. However, finding the LLE of multiple systems is computationally infeasible and not certain to improve the results.**

**Changes in Supplement:**

[Figure]

**Figure S11: Comparison between measured (Wang et al. 2014) and COSMO*therm*19 estimated Setschenow constants ($K_s$) using relative solubilities in $x_{AS}$ = 0.007.**

The authors predicted the Henry's law constant for organosulfates in water. However, in the atmosphere there are also large volume of organic phase available. What about the Henry's law constant into organic phase?

Author's response: We calculated Henry's law solubilities of the OS and IEPOX in *cis*-pinonic acid and hexanoic acid and included these values in Figure 8. The added Henry's law solubility values were also included in Table S6 in the Supplement.

**Changes in manuscript: Additionally, we calculated the infinite dilution Henry's law solubilities of all compounds in two organic solvents, hexanoic and cis-pinonic acids (see Fig. 8 and Table S6 of the Supplement). The densities of these organic acids ($\rho_{\text{hexanoic}}$ = 0.9400 g cm$^{-3}$ and $\rho_{\text{cis-pinonic}}$ = 1.0739 g cm$^{-3}$) were estimated using COSMO*therm*. The Henry's law solubilities of the monoterpene derived organosulfates are the lowest in water and the highest in *cis*-pinonic acid. The isoprene derived compounds (OS and IEPOX) are all less soluble in hexanoic acid than in water. The more oxygenated isoprene-OS-3 and -4 are also less soluble in *cis*-pinonic acid than in water, opposite to the less oxygenated isoprene-OS-1 and -2, which are the most soluble in *cis*-pinonic acid. The epoxydiols are least soluble in hexanoic acid and the most soluble in water. This means that the phase separation behavior of OS from different**

**precursors will be different in multi-phase atmospheric aerosol, leading to different OS aerosol phase state depending on the predominant precursor.**

[Figure]

**Figure 8. Comparison between infinite dilution Henry's law solubility ($H^\infty$) in water, hexanoic acid and *cis*-pinonic acid, and LLE based Henry's law solubility ($H^{LLE}$) in water. The dashed line shows 1:1 ratio between $H^\infty$ in water and the other Henry's law solubilities.**

Anonymous Referee #2

In this work, the authors have used the COSMOtherm program to calculate both properties of atmospherically relevant monoterpene and isoprene derived organosulfates (such as solubilities, activities and saturation vapor pressures). The new modeled results are important for us to better understand the atmospheric impacts and fates of organosulfates. The model simulations are carefully designed and run with strong justifications and assumptions. The paper is well written and is suitable for publication in ACP. I have two major suggestions:

1. It is understood that there are uncertainties for the model simulations. The authors have done a very nice work in explaining your model simulations. However, it would be still useful for the readers to know what would be the potential uncertainties of the modeled results (e.g. what is the possible range of the model results instead of a single value?).

Author's response: This issue was partially addressed above (see response to comment 1 of referee 1).

**Changes in manuscript: Without experimental reference data, we are not able to estimate the error for individual compounds. The error estimates are same for all studied compounds and we therefore are not showing error bars in the figures.**

2. To date, it remains unclear how the sulfate group would affect the properties of the organic compounds. It would be useful if the authors could discuss how the presence or absence of sulfate group would change the properties of an organic molecules based on their model simulations.

**Changes in manuscript: COSMO*therm* estimated Henry's law solubility has previously been reported for isoprene derived 2-methyltetrol (D'Ambro et al. 2019), which is similar to isoprene-OS-3 and -4 with the difference that the sulfate group is replaced by a hydroxy group. We calculated the Henry's law solubility of the 2-methyltetrol in water using COSMO*therm*19 and found that the compounds containing a sulfate group (isoprene-OS-3 and -4) have Henry's law solubilities that are 4 orders of magnitude higher than the compound containing only hydroxy groups (2-methyltetrol). The higher Henry's law solubility of the organosulfate, compared to the 2-methyltetrol, is caused by 5 orders of magnitude lower saturation vapor pressure and an order of magnitude higher activity coefficient at infinite dilution of the solute. Similar differences are seen between the IEPOX isomers and isoprene-OS-1 and -2, although the functional groups in isoprene-OS-1 and -2 (hydroxy and carbonyl) are different than those in IEPOX (hydroxy and epoxy). This means that the presence of sulfate in SOA and the formation of organosulfate compounds enhances SOA formation, since organosulfates are less likely to evaporate than non-sulfate organics.**

Minor comments:

Abstract, "The estimated pKa values of all the organosulfates indicate a high degree of dissociation in water, leading in turn to high dissociation corrected solubilities." Any explanation from the model simulations?

Author's response: The $pK_a$ is calculated from the energy difference of the neutral and deprotonated species (Equation 10), small energy difference explains the low $pK_a$ values. Dissociation correction is calculated using Equation 11, where low $pK_a$ leads to a high dissociation correction to the solubility.

**Changes in manuscript: The energy difference ($G_i^{anion}$-$G_i^{neutral}$) is always positive, because in a neutral solvent, a neutral compound is more favorable than a charged compound. Relatively lower anion energy (more favorable deprotonation) leads to smaller energy difference leading to lower p$K_a$.**

Line 45, "The hygroscopic properties of organosulfate containing aerosol have been measured using sodium salts of alkane sulfates (Woods III et al., 2007; Estillore et al.,2016) and limonene derived organosulfates (Hansen et al., 2015). Limonene derived organosulfate was demonstrated to lower the surface tension of aqueous solutions even more effectively than atmospherically relevant strong organic acids (Hansen et al., 2015)." Could the authors compare their model results with some of these literature data?

**Changes in manuscript: The COSMO*therm* estimated activities can be used in Köhler calculations to model the non-ideality of aqueous droplet solutions. For instance, hygroscopic growth factor (calculated as a ratio between wet and dry particle diameter) is higher for particles with lower water activity than for particles with higher water activity. From our calculation we can see that for some of the organosulfates, water activity on the organic phase is above ideality ($a_w$>$x_w$), meaning a lower water uptake compared to the organosulfates with water activities below ideality ($a_w$<$x_w$).**

Line 47, "In addition, Nguyen et al. (2014a, b) have seen indications of long-range transport of organosulfates, suggesting that organosulfates must have sufficiently low volatilities to remain in the aerosol-phase over a wide range of atmospheric conditions."This observation does not consistent with the model results? As stated in the abstract, "Based on the estimated saturation vapor pressures, the organosulfates of this study can all be categorized as semi- or low-volatile, with saturation vapor pressures 4 to 8 orders of magnitude lower than that of sulfuric acid". Do these modelled results support by the laboratory and field observations?

Author's response: Yes, the estimated saturation vapor pressures, that are 4 to 8 orders of magnitude lower than that of sulfuric acid, support the statement of OS stability in the aerosol phase. This is addressed later in the conclusions: "Calculated saturation vapor pressures are lower for organosulfates than isoprene derived dihydroxy dihydroperoxides and dihydroperoxy hydroxy aldehydes (Kurtén et al., 2018) and α-pinene derived oxidized compounds (Kurtén et al., 2016).

Based on this, organosulfates are more stable in the condensed phase than non-sulfate organic compounds. In addition, the saturation vapor pressure of $H_2SO_4$ is higher than all of the organosulfates. Due to the low p$K_a$ of all organosulfates (and $H_2SO_4$), if the aerosol contains molecules or ions capable of acting as bases, the effective vapor pressure (equilibrium vapor pressure) of OS SOA will be many orders of magnitude lower than the saturation vapor pressures. Overall, organosulfates are thus unlikely to evaporate from an aerosol in which they are formed. This means that the formation of organosulfates, and in particular the formation of their salts, can contribute significantly to increasing the SOA mass in regions with high sulfate aerosol content."

**Changes in manuscript: Not only will OS add to SOA, this SOA will also be stable over a wide range of conditions, including salinity and acidity.**

Line 80, "The non-systematic conformer generation in COSMOconf has been shown to lead to significantly different results in COSMOtherm depending on the initial geometry with molecules containing hydroxy and hydroperoxy functional groups (Kurtén et al.,2018). Based on the recommendation by Kurtén et al. (2018), we therefore used the systematic conformer sampling with the MMFF force fields in the Spartan '14 program (Wavefunction Inc., 2014)." While the organosulfates contain a sulfate group, would the recommendation made by Kurtén et al. (2018) be the best option for the organosulfates investigated in this work? More elaboration would be needed.

**Changes in manuscript: In addition to the most common carbon and oxygen atom types, MMFF force field is parametrized for the atom types of a sulfate group sulfur and oxygens (Halgren, 1996). This ensures that all unique conformers are found using the systematic sampling.**

Line 96, "All calculations were done at 298.15 K and we assume that all of the organosulfates (OS) and the isoprene epoxydiols (IEPOX) are liquids." Please elaborate why we could assume all organosulfates investigated in the work are liquids in their pure forms at 298K.

**Changes in manuscript: All calculations were done at 298.15 K. To the best of our knowledge, experimental information on the pure component phase state of most atmospherically relevant organics is not available. We therefore assume that all of the organosulfates (OS) and the isoprene epoxydiols (IEPOX) are liquid at 298.15 K. Without melting point and heat of fusion data, we are not able to accurately estimate the solubilities of solid-phase organosulfates. If the OS and IEPOX are solid at 298.15 K, the solubility results shown here are the mole fractions of the virtual liquid of the solute in the two liquid phases of a solid-liquid-liquid equilibrium. Sodium salts of the organosulfates (R−OSO$_3$Na, NaOS) are similar to sodium dodecyl sulfate (SDS) with regard to molar mass and functionality. SDS is solid at 298.15 K and we therefore assume that the NaOS are solid at this temperature.**

Line 104, "Kurtén et al. (2018) found that COSMOtherm overestimates the effect of intramolecular hydrogen bonds and recommended that only conformers containing no intramolecular hydrogen bonds should be used in saturation vapor pressure calculations. We therefore omitted all conformers containing intramolecular hydrogen bonds from the calculations of OS and IEPOX." This argument is okay. However, how this assumption would affect the model results?

**Changes in manuscript: Generally, the omission of conformers containing intramolecular H-bonds leads to lower saturation vapor pressures (Kurtén et al. 2018), higher aqueous solubility and larger deviation from ideality of activity coefficients (Hyttinen and Prisle, 2020).**

Line 268, "Compared to the binary LLE solubility, the solubility calculated as a relative solubility for monoterpene derived organosulfates is on average 3.1 times higher (1.8-5.5) using (NH4)2SO4 solutions as reference, and 2.2 times higher (1.7-2.9) using NH4HSO4 solutions." Could the authors elaborate why the relative solubility for monoterpene derived organosulfates is higher than ammonium sulfate and ammonium bisulfate?

**Changes in manuscript: Compared to the binary LLE solubility, the aqueous solubility… Based on LLE calculations, the monoterpene derived organosulfates are less soluble in the ammonium sulfate and bisulfate solutions than in pure water, which means that the ammonium salts have a salting-out effect on the OS. From the solubilities calculated as relative solubility compared to $(NH_4)_2SO_4$ and $NH_4HSO_4$ solutions, we can see that the relative solubility calculation in COSMO*therm* overestimates the salting-out effect of both ammonium salts compared to the more accurate LLE calculation. In addition, the salting-out effect of $(NH_4)_2SO_4$ is overestimated more than that of $NH_4HSO_4$. The relative solubility calculation uses the zeroth order solubility approximation, which means that the estimate is less accurate when the absolute solubility is high. The largest difference between the aqueous LLE and the solubility calculated relative to the ternary LLE are seen for the OS with the higher absolute solubilities.**

Line 286, "The organosulfates are therefore estimated to be of equivalent strength or even stronger acids than H2SO4, and thus for all practical purposes fully dissociate in near-neutral solutions and even solutions at most atmospherically relevant pH." These are important results. Any literature data to support these findings?

Author's response: Unfortunately there is no available acidity data on organosulfates. This is most likely caused by the fact that these compounds are difficult to isolate from the aerosol phase since they are present in very small quantities, just as they are highly non-trivial to synthesize in sufficient quantities in the lab. In addition, it is difficult to accurately measure $pK_a$ values of strong acids. This furthermore motivates the importance of theoretical studies on compounds such as organosulfates.

Line 288, "For all organic compounds, dissociation corrected solubilities correspond to mole fractions higher than 1. This unphysical result is likely caused by inability to accurately capture solution behavior of very strongly acidic compounds." How would this factor affect other modelled results?

**Changes in manuscript: This unphysical result is likely caused by inability of Eq. (11) to accurately capture solution behavior of very strongly acidic compounds. This equation is only used to calculate dissociation corrected solubilities and has no effect on other property calculations.**

Line 326, "With O:C ratios of the monoterpene and isoprene derived organosulfates in the ranges 0.5–0.7 and 1.2–1.4, respectively, these results are consistent with the present work. On the other hand, in experiments of OH oxidized α-pinene and water system (Ham et al., 2019) only a single organic-rich phase was observed, whereas LLPS was seen between water and ozone oxidized α-pinene products (Ham et al.,2019) or OH oxidized isoprene products (Rastak et al., 2017)." For organosulfates, 4 oxygen atoms are associated from the sulfate group. Would this affect how we interpret the O:C ratio?

**Changes in manuscript: There are small differences in the partial charges of the oxygen atoms associated to a sulfate group compared to those associated to a carboxylic acid group (see Section S3 of the Supplement for a comparison of the σ-profiles) that may influence the O:C ratio of organosulfates required for LLPS. From our results we can also see that other structural factors further affect the thermodynamic properties, in addition to the O:C ratio or the types of functional groups.**

**Changes in Supplement: A sulfate group contains two double bonded oxygen atoms, one OH and one ether type oxygen. We have plotted the σ-profiles (Figure S9) and -potentials (Figure S10) of the SO$_2$H fragment of a sulfate group (corresponding to a carboxylic acid with the carbon replaced by sulfur), the whole sulfate group (OSO$_3$H) and a carboxylic acid (COOH). Comparing the σ-surfaces of the sulfate group and the carboxylic acid group shows that the partial charges of the sulfate group (oxygens and hydrogen) are more positive than the charges of a carboxylic acid (see Figure S9 of the Supplement). The partial negative charge of the oxygen atoms of the sulfate group are slightly less negative but the charge covers a larger surface area than in the carboxylic acid oxygens. From the σ-potentials we can see that carboxylic acid is able to act as both a hydrogen bond acceptor and a donor, while sulfate is characterized only as a hydrogen bond donor. However, sulfate is a stronger hydrogen bond donor than carboxylic acid and carboxylic acid is a weaker H-bond acceptor than donor. Based on the σ-potentials, the negative partial charge of the sulfate group is too weak to be counted as H-bond acceptor in the σ-potential.**

[Figure]

Figure S5: σ-profiles of the carboxylic acid group, sulfate group and SO₂H of the sulfate group (corresponding to a carboxylic acid group) of α-pinene-OS-5_c22. Negative σ values (screening charge density) indicate a positive partial charge and positive σ values negative partial charge.

[Figure]

Figure S6: σ-potentials of the carboxylic acid group, sulfate group and SO₂H of the sulfate group (corresponding to a carboxylic acid group) of α-pinene-OS-5_c22. Negative μ(σ) values indicate favorable interaction with partial charges corresponding to the σ values on the x-axis. For example, all of the groups shown here are able to act as H-bond donors (interaction

**with negative partial charges).**

Line 375, "We calculated saturation vapor pressures for the neutral organic compounds at 298.15 K (Table 1). Comparing the studied organosulfate compounds based on their functional groups, those containing carboxylic acid groups, i.e., α-pinene-OS-5 and α-pinene-OS-6, have the lowest saturation vapor pressures." How the presence of sulfate group would affect the saturation vapor pressure of the organic compounds?

Line 403, "Among the studied organics, Henry's law solubility is the highest for monoterpene and isoprene derived organosulfates containing the highest number of oxygen atoms and the lowest for methyl bisulfate and the IEPOX isomers." Also, how the presence of sulfate group would affect the Henry's law solubility of the organic compounds?

Author's response: We have added a comparison after the Henry's law results (see response to comment 2 of referee 2 above).

**New references:**

D'Ambro, E. L., Schobesberger, S., Gaston, C. J., Lopez-Hilfiker, F. D., Lee, B. H., Liu, J., Zelenyuk, A., Bell, D., Cappa, C. D., Helgestad, T., Li, Z., Guenther, A., Wang, J., Wise, M., Caylor, R., Surratt, J. D., Riedel, T., Hyttinen, N., Salo, V.-T., Hasan, G., Kurtén, T., Shilling, J. E., and Thornton, J. A.: Chamber-based insights into the factors controlling epoxydiol (IEPOX) secondary organic aerosol (SOA) yield, composition, and volatility, Atmos. Chem. Phys., 19, 11 253–11 265, https://doi.org/10.5194/acp-19-11253-2019, 2019.

Halgren, T. A.: Merck molecular force field. I. Basis, form, scope, parameterization, and performance of MMFF94, J. Comp. Chem., 17, 490–519, https://doi.org/10.1002/(SICI)1096-987X(199604)17:5/6<490::AID-JCC1>3.0.CO;2-P, 1996.

Hyttinen, N. and Prisle, N. L.: Improving solubility and activity estimates of multifunctional atmospheric organics by selecting conformers in COSMO*therm*, submitted, 2020.

Klamt, A., Eckert, F., Reinisch, J., and Wichmann, K.: Prediction of cyclohexane-water distribution coefficients with COSMO-RS on the SAMPL5 data set, J. Comput.-Aided Mol. Des., 30, 959–967, https://doi.org/10.1007/s10822-016-9927-y, 2016.

Schröder, B., Fulem, M., and Martins, M. A. R.: Vapor pressure predictions of multi-functional oxygen-containing organic compounds with COSMO-RS, Atmospheric Environment, 133, 135–144, https://doi.org/10.1016/j.atmosenv.2016.03.036, 2016.